# Mitotic DNA synthesis in response to replication stress requires the sequential action of DNA polymerases zeta and delta in human cells

Wei Wu[1,4,6] ✉, Szymon A. Barwacz [1,6], Rahul Bhowmick[1,5], Katrine Lundgaard [1], Marisa M. Gonçalves Dinis [1], Malgorzata Clausen[1], Masato T. Kanemaki [2,3] & Ying Liu [1] ✉

Oncogene activation creates DNA replication stress (RS) in cancer cells, which can generate under-replicated DNA regions (UDRs) that persist until cells enter mitosis. UDRs also have the potential to generate DNA bridges in anaphase cells or micronuclei in the daughter cells, which could promote genomic instability. To suppress such damaging changes to the genome, human cells have developed a strategy to conduct 'unscheduled' DNA synthesis in mitosis (termed MiDAS) that serves to rescue under-replicated loci. Previous studies have shown that MiDAS proceeds via a POLD3-dependent pathway that shows some features of break-induced replication. Here, we define how human cells utilize both DNA gap filling (REV1 and Pol ζ) and replicative (Pol δ) DNA polymerases to complete genome duplication following a perturbed S-phase. We present evidence for the existence of a polymerase-switch during MiDAS that is required for new DNA synthesis at UDRs. Moreover, we reveal that, upon oncogene activation, cancer cell survival is significantly compromised when REV1 is depleted, suggesting that REV1 inhibition might be a feasible approach for the treatment of some human cancers.

Tumorigenesis is a Darwinian evolutionary process where natural selection acts upon spontaneously generated genetic changes in somatic cells. These changes enable the activation of oncogenes or the loss of function of tumor suppressor genes, which allows cancer cells to gain critical phenotypic or survival advantages. Recently, it was reported that, on average, a cancer genome contains 4 or 5 driver mutations[1]. Amongst these mutations, at least one always leads to the activation of an oncogene.

Oncogene activation is a major source of DNA replication stress (RS) in tumors. RS is characterized by increased replication fork stalling, an accumulation of collapsed replication forks, and the activation of a DNA damage response (DDR) in the early stages of cancer development[2–5]. Under severe RS, a DDR mediated by the ATM- and Rad3-related (ATR) kinase and its effector kinase, checkpoint kinase 1 (Chk1), can trigger senescence or apoptosis in untransformed cells[6,7]. Only when additional genetic or

[1]Center for Chromosome Stability, Department of Cellular and Molecular Medicine, University of Copenhagen, Blegdamsvej 3B, 2200 Copenhagen N, Denmark. [2]Department of Chromosome Science, National Institute of Genetics, Research Organization of Information and Systems (ROIS), Yata 1111, Mishima, Shizuoka 411-8540, Japan. [3]Department of Genetics, The Graduate University for Advanced Studies (SOKENDAI), Yata 1111, Mishima, Shizuoka 411-8540, Japan. [4]Present address: Zhejiang Provincial Key Laboratory of Pancreatic Disease, First Affiliated Hospital of Zhejiang University, Hangzhou, Zhejiang 310003, China. [5]Present address: Department of Biochemistry, Vanderbilt University School of Medicine, Nashville, TN 37232, USA. [6]These authors contributed equally: Wei Wu, Szymon A. Barwacz. ✉e-mail: 308870715@qq.com; ying@sund.ku.dk

epigenetic changes occur (e.g. loss of the p53 pathway), which lead to the DDR pathway being compromised, does tumorigenesis become possible[8,9].

A set of loci in human cells, termed common fragile sites (CFSs), are known to be associated with translocation breakpoints[10], deletions and translocations[11,12] in cancer cells, as well as with viral integration sites[13–16]. CFSs are genomic regions that are deemed to be 'difficult-to-replicate' and display gaps or breaks visible on condensed metaphase chromosomes when cells are exposed during S-phase to a low dose (0.2–0.6 μM) of the B-family DNA polymerase inhibitor, aphidicolin, (APH), which generates RS[17]. Several genomic features have been suggested to explain why CFSs are difficult-to-replicate, particularly when cells face RS (e.g. oncogene activation or APH treatment). First, CFSs frequently contain very large genes, which can cause collisions to arise between the replication and transcription machineries, and trigger RNA-DNA hybrid (R-loop) formation[18,19]. Second, some CFSs are associated with AT-rich sequences and contain AT microsatellite repeats that are prone to form replication-blocking DNA secondary structures[20]. Third, CFSs are often characterized not only by being late replicating, but also by possessing a low density of replication origins, which results in long-traveling replication forks that are prone to be unstable[21]. Therefore, when cells are challenged by RS, CFSs might remain under-replicated by the end of the G2 phase. If these under-replicated regions escape detection by the G2/M checkpoint, cells might proceed into M-phase even though some CFS regions remain under-replicated, leading to a local delay in DNA condensation, which manifests as fragility at these loci[22,23].

Several pathways exist in human cells to ameliorate the deleterious effects of RS. When cells are subjected to RS, the ATR signaling pathway is activated, and this leads to the recruitment of the FANCD2-FANCI complex and DNA polymerase η (Pol η) to CFSs to promote their stability[24–27]. Despite this, some CFSs remain under-replicated by the end of interphase, as evidenced by the persistence of FANCD2/I foci at the site of CFS loci on metaphase chromosomes[28]. Similarly, SLX4 co-localizes with FANCD2 throughout G2 and mitosis[29,30]. A study in DT40 cells revealed that TopBP1 promotes the recruitment of SLX4 to FANCD2 foci[31], although this has not been verified in human cells to our knowledge. Upon mitotic entry, due to the action of the CDK1 and PLK1 kinases, the structure-specific endonuclease (SSE), MUS81-EME1, associates with SLX4 and promotes not only CFS expression[30,32–35], but also an atypical form of DNA synthesis that occurs in mitosis called MiDAS[30]. If MiDAS cannot proceed, the sister chromatids can remain entangled, which leads to the formation of bulky chromatin bridges or ultra-fine DNA bridges (UFBs) in anaphase[28,30]. In those UFBs that derive from CFSs, FANCD2 localizes on each chromatid at the termini of the bridges[28]. Because of its localization at CFSs from late S-phase until late mitosis, FANCD2 is widely used as a surrogate marker of the location of CFSs that have undergone perturbed replication[28,30]. In the following G1 phase, any unresolved or aberrantly processed DNA bridges can form either micronuclei or be marked as problematic DNA regions by the presence of so called 53BP1 bodies[36].

The precise mechanism by which MiDAS occurs is not known, although it clearly requires multiple steps and involves the cooperation of numerous proteins. To date, the following proteins are proposed to contribute to one of four key steps in the process, although the details remain to be defined: i) TRAIP, a ubiquitin E3 ligase, drives replisome disassembly in response to any remaining stalled replication forks as cells enter mitosis[37]; ii) the SLX4-associated SSE complex cleaves the 'exposed' DNA at the stalled fork[30,38]; iii) RTEL1 (regulator of telomere elongation helicase) then potentially unwinds atypical DNA structures[39]; iv) RAD52, a protein that can anneal to ssDNA and help with HR activities in human cells (reviewed in[40,41]), and POLD3, the

human homologue of yeast Pol32, are both essential for promoting the DNA synthesis that occurs during MiDAS[30,40]. It has been proposed that this DNA synthesis might proceed via a pathway resembling the break-induced replication (BIR) pathway defined in yeast[42]. However, it remains to be clarified which DNA polymerase(s) functions in MiDAS. Considering that POLD3 is a subunit of the replicative DNA polymerase δ (Pol δ)[43] and one of the key translesion (TLS) polymerases, Pol ζ[44], we hypothesized that either or both of Pol ζ and Pol δ might play a role in MiDAS.

TLS polymerases are a group of enzymes that are specialized in promoting the bypass of DNA damage sites. In human cells, there are at least 11 TLS enzymes that operate in different DNA damage pathways (reviewed in[45]). One of these, Pol ζ, is composed of two main subunits: Rev3 (catalytic unit) and Rev7 (accessory unit)[46]. This enzyme is believed to be the main polymerase that can perform extension from a misaligned primer in an error-free or error-prone way (reviewed in[45]). More recently, it was revealed that human Pol ζ can contain 2 subunits in addition to REV3 and REV7, and this holoenzyme is referred to as Pol ζ4[44]. These two additional subunits are POLD2 and POLD3, both of which are also subunits of human Pol δ. Interestingly, Pol ζ4 is more efficient in bypassing bulky lesions than is Pol ζ2 (the complex of REV3 and REV7)[44]. Therefore, it has been suggested that, during some forms of DNA synthesis, Pol ζ and Pol δ might switch roles via an exchange of their shared subunits[47]. Of possible significance, both Pol ζ and REV1, a protein acting as a scaffolding factor for other TLS Pols[48], facilitate a specific type of BIR called microhomology-mediated BIR (mmBIR)[49]. In human cells, REV1 has been shown to cooperate with Pol ζ to promote TLS[50], as well as to interact with POLD3/Pol32 in human and yeast cells, respectively[51,52]. Moreover, it was demonstrated that, in addition to their function in S-phase, REV1 and Pol ζ operate in G2 to facilitate replication of damaged DNA induced by UV-radiation[53]. It is plausible, therefore, that REV1 would also play a role in MiDAS.

In this study, we set out to define whether there is a requirement for POLD1 (the catalytic subunit of human Pol δ), REV1, REV3 or REV7 in MiDAS. Interestingly, our data demonstrate that POLD1, REV3 and REV1 are essential for MiDAS, while REV7 is not. Consistent with this, we demonstrate that REV1 and POLD1 co-localize with FANCD2 in mitosis following RS, suggesting they both indeed play a role in the 'rescue' of under-replicated regions. Mechanistically, we show that RAD18-mediates PCNA monoubiquitylation at residue K164R, which is crucial for the recruitment of REV1 to chromatin to promote MiDAS. We also show that the catalytic domain of REV3 and the REV1-interacting region (RIR) of POLD3 are crucial for MiDAS. Together, our data suggest that MiDAS involves an unusual DNA replication process involving the function of both TLS and Pol δ polymerases. We speculate that there might be a 'switch' from Pol ζ to Pol δ during this process via the action of the POLD3 subunit.

## Results

### POLD1, REV3, and REV1 facilitate MiDAS

To address whether any of the POLD3-associated DNA polymerases play a role in MiDAS, we first asked whether the catalytic subunit of Pol δ, POLD1, is critical for MiDAS. Because POLD1 is essential for cell survival, we established an HCT116 cell line with stable expression of POLD1 tagged with an mAID-Clover (mAC) degron (hereafter the 'HCT116-POLD1-AID2 degron cell line'), which allows a rapid and efficient degradation of POLD1 in early mitosis upon treatment of these cells with 5-Ph-IAA[54] (Fig. 1a–c). We then performed MiDAS analysis (EdU incorporation in early mitosis) in these cells following an established protocol[55]. Our results showed that, following low dose APH treatment in S-phase, and depletion of POLD1 at the G2/M boundary (Fig. 1d, e and Supplementary Fig. 1), MiDAS was largely abolished when the cells were released into mitosis (Fig. 1f, g). These data are consistent with the observation that MiDAS is blocked by treatment of cells with a high dose of APH (2 μM) in mitosis[30]. To investigate

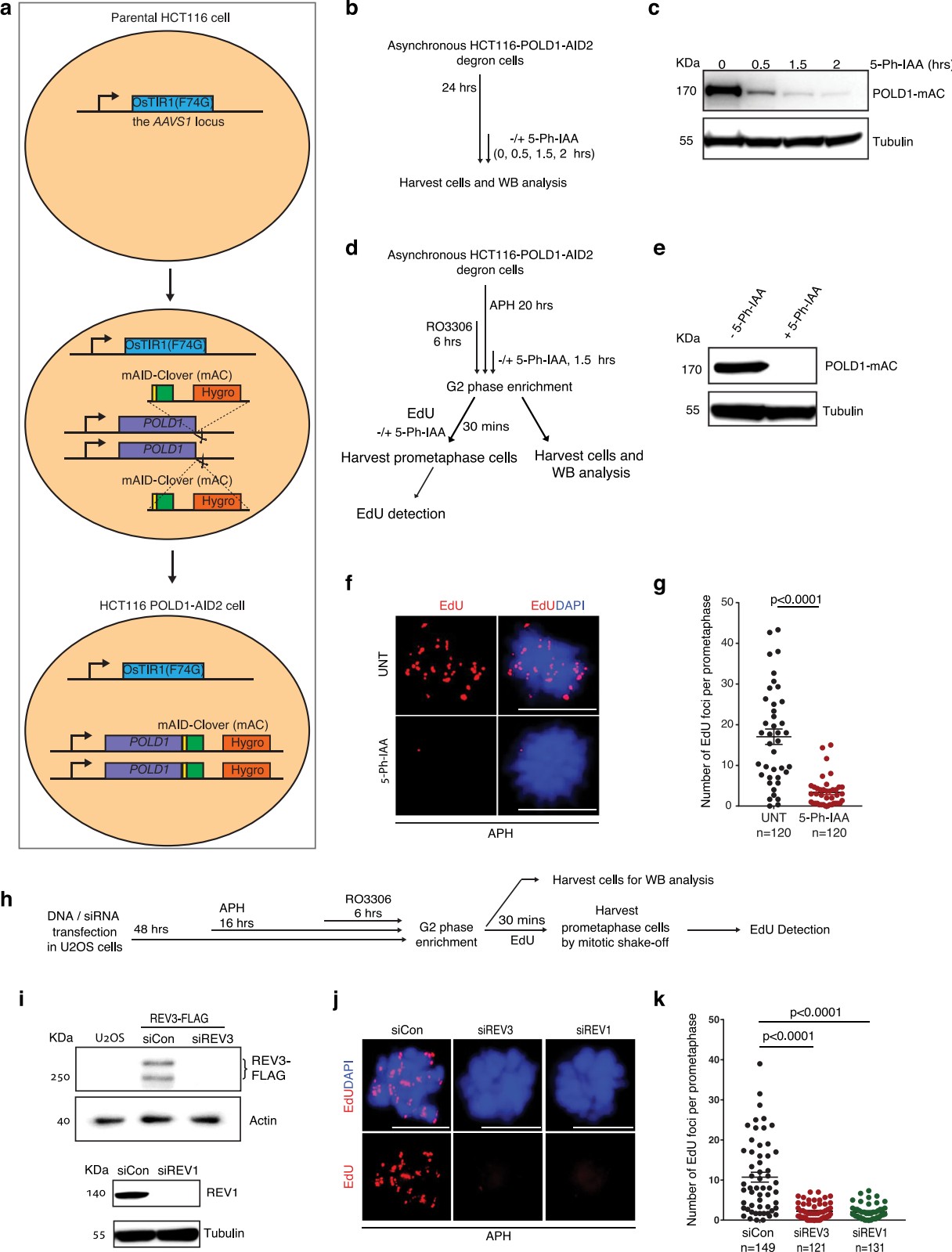

whether other POLD3-associated polymerases might also promote MiDAS, we analyzed mitotic EdU incorporation in U2OS cells depleted of the catalytic subunit of Pol ζ, REV3, or the associated REV1 protein (Fig. 1h–k). We observed that MiDAS was also largely abolished when either REV3 or REV1 was depleted, indicating that TLS polymerases also play a role in MiDAS. In addition, we could confirm that REV1 and REV3 are essential for MiDAS in other cancer cell lines; namely, HeLa,

HCT116, and the HCT116-POLD1-AID2 cell line that we have established in this study (Supplementary Fig. 2).

To verify the function of REV3 in MiDAS, we next asked whether the catalytic domain of REV3 is crucial in this process. For this, we created a mutant REV3-FLAG plasmid based on a published REV3-FLAG construct[56]. In our modified construct, the REV3 catalytic 'YGDTDS' motif was changed to a catalytically-dead 'YGATAS' motif[57]. We then

**Fig. 1 | POLD1, REV3 and REV1 are essential for MiDAS. a** Workflow for establishing an HCT116-POLD1-AID2 cell line expressing only POLD1 that is tagged with OsTIR1(F74G) (mAC). **b** Experimental workflow for assessing POLD1 protein degradation following different periods of incubation with 5-Ph-IAA. **c** Western blot (WB) analysis of POLD1-mAc in asynchronous cells treated as shown in panel **b**. β-tubulin was a loading control. **d** Experimental workflow for analysis of MiDAS in prometaphase HCT116-POLD1-AID2 cells following APH treatment, enrichment of G2 cells by RO3306, and POLD1-mAC degradation. POLD1-mAC depletion was induced by treating cells with 5-Ph-IAA for 2 h. **e** WB analysis of POLD1-mAC (detected with a POLD1 antibody) following 2 h 5-Ph-IAA treatment. β-tubulin was used as a loading control. Representative images (**f**) and quantification (**g**) of MiDAS foci (labeled with EdU; red) in prometaphase cells treated as shown in panel **d**. **h** Experimental workflow for U2OS cell synchronization and analysis of MiDAS in prometaphase cells following REV3 or REV1 depletion. In **b** and **h**, before being incubated with EdU for 30 min (with or without 5-Ph-IAA), cells were rinsed three times with pre-warmed, drug-free culture medium within 5 min. **i** Top, WB analysis of REV3FLAG following transfection of a REV3FLAG expressing plasmid together with control or REV3 siRNAs. Flag antibody was used to detect flag-REV3, since a reliable REV3 antibody is not available. Actin was used as a loading control. Protein sample from untransfected U2OS cells was used as a negative control. Lower, WB analysis of REV1 after transfecting cells with control or REV1 siRNAs. β-tubulin was used as a loading control. Representative images (**j**) and quantification (**k**) of MiDAS foci (labeled with EdU; red) in prometaphase cells treated as shown in panel **h**. DNA was stained with DAPI (blue). Each data point in charts of **g** and **k** is means of three independent experiments and plotted with Prism (*n* = number of cells analyzed in each condition in three independent experiments). Error bars represent SEM. *P* values were calculated using a two-tailed non-parametric Mann–Whitney test. Scale bars, 10 μm. Hr hour, min minute.

performed MiDAS analysis in U2OS cells with ectopic expression of either wild type REV3-FLAG or mutant REV3-ATA-FLAG while the endogenous *REV3* gene was silenced by an siRNA targeting the 3′UTR. Our results indicate that cells transfected with wild type REV3-FLAG display significantly more MiDAS than the cells transfected with the REV3-ATA-FLAG mutant (Supplementary Fig. 3), indicating that the catalytic function of REV3 is essential for MiDAS.

To further validate the above findings, we addressed whether REV1 could localize to mitotic cells following RS in S-phase. Therefore, we treated U2OS cells with low dose APH for 16 h, and performed immunofluorescence (IF) analysis in asynchronous, prophase and prometaphase cells with antibodies against FANCD2, POLD1 or REV1. This was conducted to examine whether POLD1 and/or REV1 are recruited to loci undergoing replication fork perturbation (mainly CFSs) that are marked by FANCD2 foci (Supplementary Fig. 4a). Indeed, we observed a significantly increased degree of co-localization of both POLD1 and REV1 with FANCD2 in asynchronous cells, prophase cells, and prometaphase cells upon APH treatment (Supplementary Fig. 4b–e). Previous studies have also implicated one of the TLS polymerases, Pol η (POLH), in the maintenance of CFS stability[24]. Pol η was also reported to cooperate with Pol ζ4 to bypass cisplatin lesions by inserting dCTP opposite the 3′ guanine[44]. Therefore, we addressed whether this polymerase might also affect MiDAS. However, we observed that depletion of Pol η did not affect the frequency of MiDAS (Supplementary Fig. 5). We conclude, therefore, that POLD1, REV1 and REV3, but not POL η, contribute to MiDAS.

To address whether inactivation of Pol ζ or REV1 could cause genomic instability similar to that caused by inactivation of POLD3[30], we used siRNAs to deplete REV3 or REV1, and then analyzed DNA bridges in anaphase cells following treatment of cells with low dose APH (Supplementary Fig. 6a). Following the depletion of either REV3 or REV1, we observed a significant increase of both chromatin bridges and PICH-coated UFBs (an indication of UFBs comprising predominantly dsDNA) in U2OS cells (Supplementary Fig. 6b–e). In addition, newly born G1 cells from the REV3- or REV1-depleted cells showed a significant increase in the frequency of both micronuclei and 53BP1 bodies (Supplementary Fig. 6f–h).

## RAD18 mediated PCNA monoubiquitylation recruits REV1 to promote MiDAS

Considering that Pol δ is a highly processive polymerase[58], while Pol ζ generally acts to synthesize only short stretches of DNA[44], we considered the possibility that MiDAS might be initiated by a TLS polymerase, which then hands over the task of performing extended DNA synthesis to Pol δ. Therefore, we analyzed how TLS polymerases are recruited to MiDAS loci. It was shown previously that, in response to DNA damage, PCNA becomes monoubiquitylated on lysine-164 (K164), which is essential for the recruitment of TLS polymerases[59]. To test the possibility that this process occurs during MiDAS, we first assessed whether PCNA is present at FANCD2 positive loci in mitosis when cells are challenged with RS. We observed clear co-localization of FANCD2 with PCNA in asynchronous, prophase, and prometaphase cells following low dose APH treatment (Supplementary Fig. 7a–c). We also confirmed that PCNA is essential for MiDAS (Supplementary Fig. 7d–g). We therefore analyzed whether the monoubiquitylation of PCNA on K164 is a critical feature of MiDAS. For this, we employed an U2OS cell line with stable expression of a PCNA[K164R] mutant that cannot be ubiquitylated (Fig. 2a, b). Upon depletion of the endogenous (wild-type) PCNA in these cells, we observed that both MiDAS and the recruitment of REV1 to chromatin in mitotic cells were compromised (Fig. 2c–f). Furthermore, because the K164 residue of PCNA site can also be SUMOylated[60], we tested whether general inhibition of SUMOylation might affect MiDAS. However, we observed that inhibition of SUMOylation by depletion of UBC9 did not affect MiDAS (Fig. 2g–j). Taken together, these results indicate that monoubiquitylation of PCNA at K164 is required for REV1 to promote MiDAS.

Next, we sought to identify the E3 ubiquitin ligase that is responsible for modification of PCNA following low dose APH treatment. So far, there are two well characterized E3 ubiquitin ligases that can conjugate ubiquitin to PCNA; namely, RAD18 and CDT2. It was reported previously that RAD18 acts following DNA damage[61], while CDT2 ubiquitylates PCNA under unstressed conditions[62]. To analyze this in cells treated with APH, we depleted these two ligases either alone or in combination, and then analyzed PCNA monoubiquitylation by Western blot analysis (Fig. 3a). We observed that, following APH induced RS, RAD18 depletion, but not CDT2 depletion, abolished PCNA monoubiquitylation. On the contrary, CDT2 depletion induced a strong induction of PCNA monoubiquitylation that could be suppressed by co-depletion of RAD18. These data indicate that RAD18, and not CDT2, is the E3 ubiquitin ligase responsible for PCNA monoubiquitylation following APH treatment (Fig. 3b, c). Next, we tested whether RAD18 mediated PCNA monoubiquitylation plays a role in MiDAS. Indeed, consistent with the finding that PCNA ubiquitylation is required for MiDAS, RAD18 depletion also largely abolished MiDAS. In contrast, CDT2 depletion led to a significant induction in the level of MiDAS, which could be suppressed by co-depletion of RAD18 (Fig. 3d–f). It is plausible that, when CDT2-depleted cells are treated with APH, there is an increase in the degree of RS due the loss of the 'housekeeping' function of CDT2 in ubiquitylation cascades. Thus, we conclude that, in genomic regions where replication cannot be completed due to RS, PCNA is monoubiquitylated by RAD18 on K164, which then recruits REV1 to these regions to facilitate MiDAS.

## POLD1 acts downstream of TLS in MiDAS

Next, we set out to determine whether Pol ζ functions upstream or downstream of Pol δ in the MiDAS pathway. If, as proposed previously[30,40], MiDAS requires DNA strand invasion followed by extensive DNA synthesis, we reasoned that factors acting upstream or very early in this HR-based process would suppress the formation of RPA-coated UFBs in anaphase that represent recombination

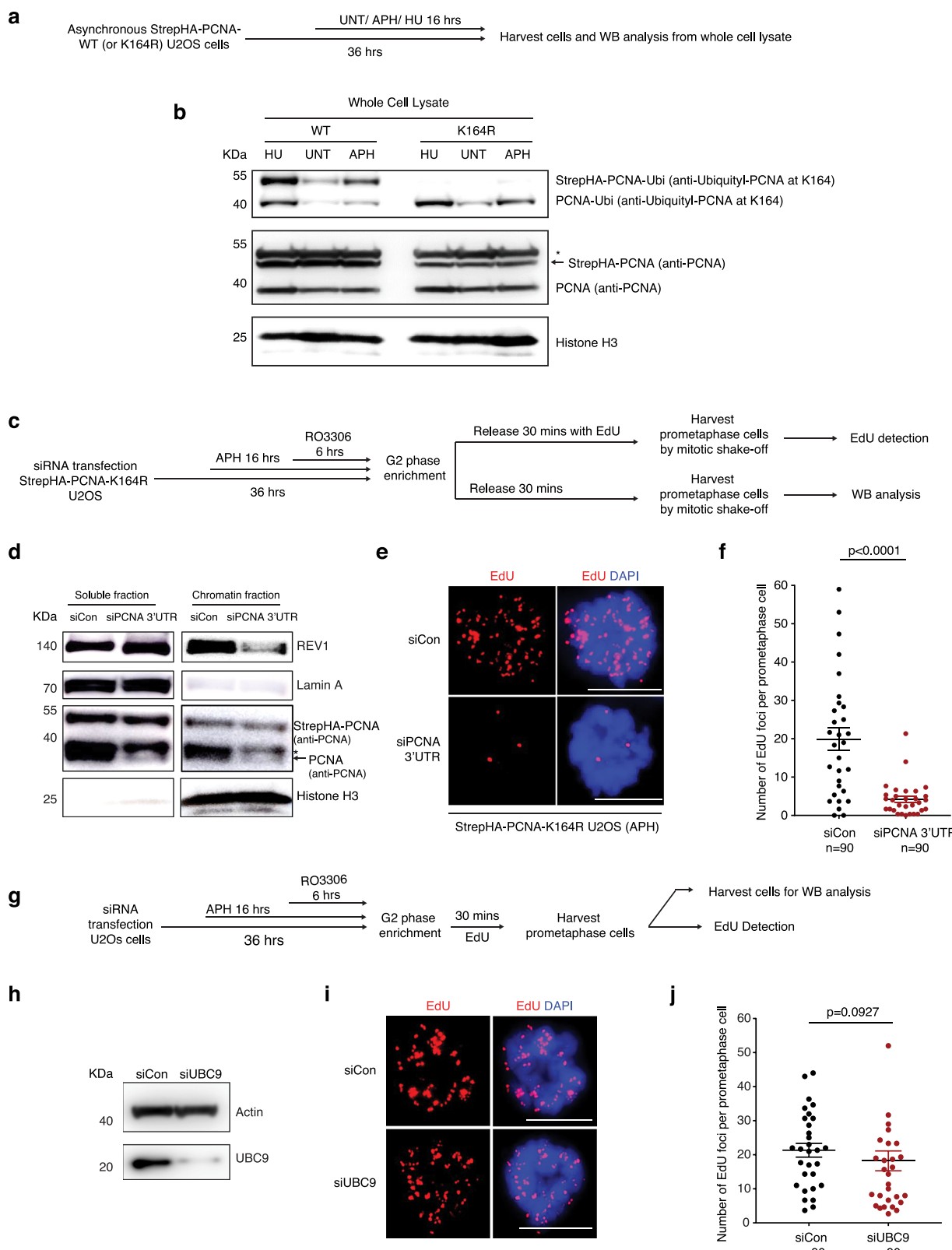

intermediates[63–65] but retain PICH-associated UFBs that define late replication intermediates. This is because the process would be blocked prior to strand invasion. In contrast, loss of factors acting late and downstream of strand invasion and D-loop formation would lead to an accumulation of unprocessed HR intermediates, i.e. RPA-coated UFBs in anaphase. To test this general hypothesis, we analyzed HCT116-POLD1-AID2 degron cells and performed immunofluorescence

analysis in anaphase cells following treatment of the cells with low dose APH in interphase (Fig. 4a). We observed that, when POLD1 was depleted in early mitosis, there was a large increase in the frequency of RPA-coated, but not PICH-coated, UFBs in anaphase (Fig. 4b–e). In contrast, when REV3 was depleted, the frequency of RPA-coated UFBs was increased to only a very modest extent, but levels of PICH-coated UFBs were increased substantially (Fig. 4b–e), which is in agreement

**Fig. 2 | PCNA monoubiquitylation is required for MiDAS. a** Experimental workflow of Western blot (WB) with StrepHA-PCNA WT or K164R U2OS cells. **b** WB analysis of asynchronous StrepHA-PCNA WT or K164R U2OS cells with antibodies against PCNA monoubiquitylation at Lys-164, unmodified PCNA, or Histone H3. * denotes a non-specific band detected by the PCNA antibody. Strep-HA-PCNA is indicated by a black arrow. **c** Experimental workflow for analysis of MiDAS in prometaphase StrepHA-PCNA K164R U2OS cells following endogenous PCNA depletion and APH treatment, and enrichment of G2 cells with RO3306 treatment. **d** WB analysis of the soluble and chromatin bound endogenous PCNA, StrepHA-PCNA(K164R) and REV1 in the StrepHA-PCNA K164R cells, after cells were transfected with control siRNA or PCNA 3′UTR siRNA. * denotes a non-specific band detected by the PCNA antibody. PCNA is indicated by the black arrow. In panels **b** and **d**, Lamin A was used as a loading control for soluble proteins, and Histone H3 was used as a loading control for chromatin-bound proteins. Representative images

(**e**) and quantification (**f**) of MiDAS foci (labeled with EdU; red) in prometaphase cells treated as shown in panel **c**. DNA was stained with DAPI (blue). **g** Experimental workflow for analysis of MiDAS in prometaphase U2OS cells following siRNA treatment of UBC9, APH treatment (0.4 μM), and enrichment of G2 cells with RO3306. In **c** and **g**, before being incubated with EdU for 30 min, cells were rinsed three times with pre-warmed, drug-free medium within 5 min. **h** WB analysis of UBC9 after transfecting cells with control or UBC9 siRNAs. Actin was used as a loading control. Representative images (**i**) and quantification (**j**) of MiDAS foci (labeled with EdU; red) in prometaphase cells treated as shown in panel **c**. DNA was stained with DAPI (blue). minute. Each data point in charts of **f** and **j** is means of three independent experiments and plotted with Prism (*n* = number of cells analyzed in each condition in three experiments). Error bars represent SEM. *P* values were calculated using a two-tailed non-parametric Mann–Whitney test. Scale bars, 10 μm. Hr hour, min minutes.

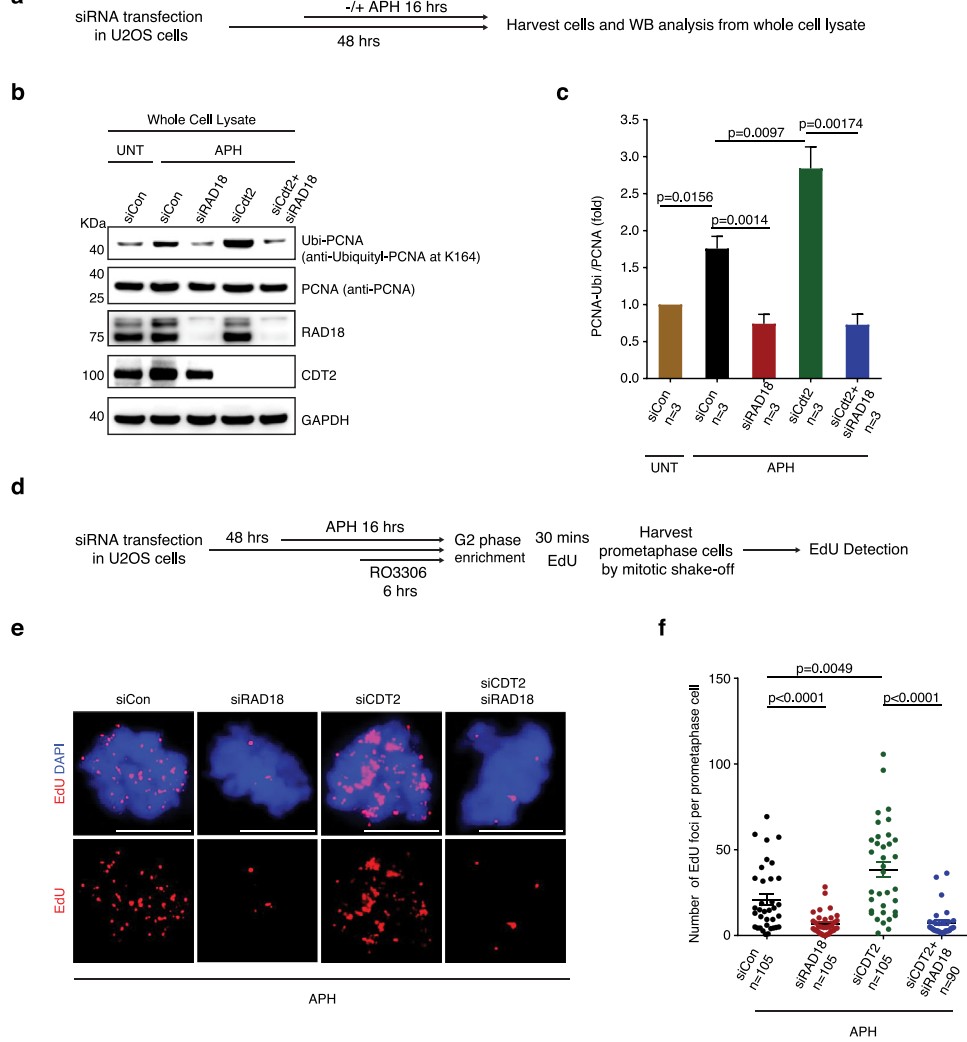

**Fig. 3 | RAD18, but not Cdt2, mediates PCNA monoubiquitylation to promote MiDAS. a** Experimental workflow of Western blot (WB) analysis of asynchronous U2OS cells following different treatment. Representative images (**b**) and quantification (**c**) WB analysis of PCNA monoubiquitylation and other protein expression in whole cell lysate from asynchronous U2OS cells with antibodies specific for PCNA monoubiquitylation at Lys-164, total PCNA, RAD18 or Cdt2. GAPDH was used as a loading control. Data of each bar are means of independent experiments. Error bars represent SEM. *P* values were calculated using a two-tailed Student's *t*-test (*n* = 3 biological replicates). **d** Experimental workflow for analysis of MiDAS in prometaphase U2OS cells following RAD18 depletion, Cdt2 depletion or co-depletion of

RAD18 and Cdt2, APH treatment, and enrichment of G2 cells with RO3306. Before being incubated with EdU for 30 min, cells were rinsed three times with pre-warmed, drug-free culture medium within 5 min. Representative images (**e**) and quantification (**f**) of MiDAS foci (labeled with EdU; red) in prometaphase cells treated as shown in panel **d**. DNA was stained with DAPI (blue). Each data point in chart **f** is means of three independent experiments and plotted with Prism (*n*= number of cells analyzed in each condition in three experiments). Error bars represent SEM. *P* values were calculated using a two-tailed non-parametric Mann–Whitney test. Scale bars, 10 μm. Hr hour, min minute.

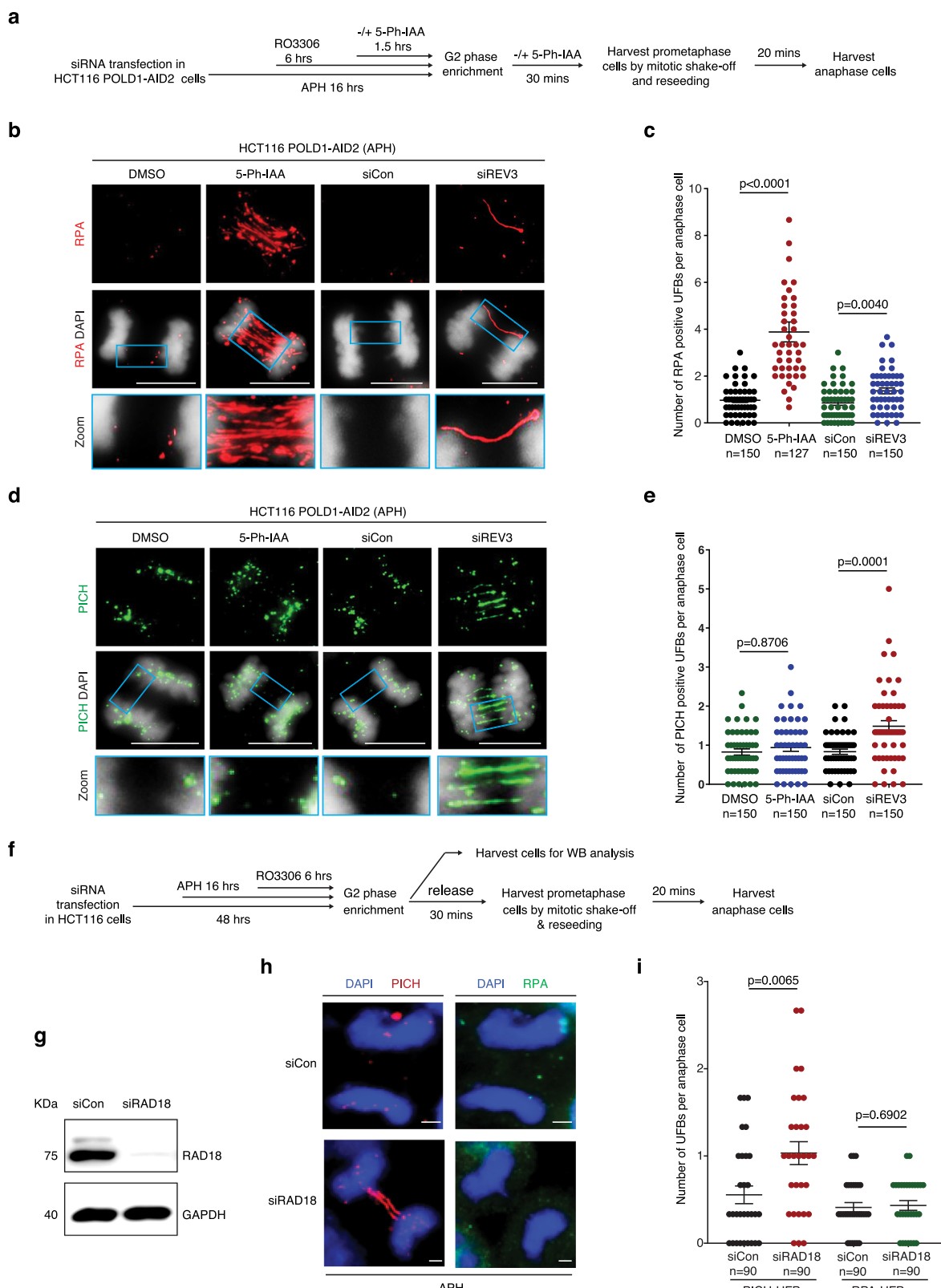

with previous findings[30,66,67]. These data are consistent with the notion that Pol δ might act downstream of Pol ζ during MiDAS. Furthermore, because RAD18 ubiquitylates PCNA in response to APH treatment, and plays a direct role in MiDAS, we quantified the frequency of PICH- and RPA-coated UFBs following RAD18 depletion (Fig. 4f). We observed that there was a significantly increased level of PICH-coated UFBs, but not RPA-coated UFBs, in RAD18-depleted cells. This supports the

notion that Pol ζ functions upstream of Pol δ in the MiDAS pathway (Fig. 4g–i).

To address this issue more rigorously, we analyzed the order in which TLS or Pol δ polymerases are recruited to chromatin in prometaphase when MiDAS is required. For this, we quantified the level of chromatin-bound REV1, REV7, POLD1 and POLD3 in prometaphase cells in which REV1, REV3, or POLD3 had been depleted using siRNAs. It

**Fig. 4 | POLD1, but not REV3, depletion induces extensive RPA-coated UFB formation. a** Experimental workflow for anaphase cell synchronization and analysis of RPA-coated UFBs in HCT116-POLD1-AID2 cells (HCT116 cells expressing OsTIR1, F74G) following REV3 depletion using siRNAs or POLD1 depletion by 5-Ph-IAA treatment. Representative immunofluorescence images (**b**) and quantification (**c**) of RPA-coated UFBs (red) in anaphase cells treated as shown in panel **a**. DNA was stained with DAPI (blue). Representative immunofluorescence images (**d**) and quantification (**e**) of PICH-coated UFBs (green) in anaphase cells treated as shown in panel **a**. DNA was stained with DAPI (blue). **f** Experimental workflow for analysis of PICH- or RPA-coated UFBs in HCT116 cells following RAD18 depletion using siRNAs.

In **a** and **f**, before being incubated with fresh medium with or without 5-Ph-IAA for 30 min, cells were rinsed three times with pre-warmed, drug-free culture medium within 5 min. **g** WB analysis of RAD18 after transfecting HCT116 cells with control or RAD18 siRNAs. GAPDH was used as a loading control. Representative immunofluorescence images (**h**) and quantification (**i**) of PICH-UFBs (red) or RPA-UFBs (green) in anaphase HCT116 cells. DNA was stained with DAPI (blue). Each data point in charts of **c**, **e**, and **i** is means of three independent experiments and plotted with Prism (*n* = number of cells analyzed in each condition in three experiments). Error bars represent SEM. *P* values were calculated using a two-tailed non-parametric Mann–Whitney test. Scale bars, 10 μm. Hr hour, min minute.

should be noted that, due to the general low expression of REV3, we were unable to assess the chromatin-bound fraction of REV3. We observed that REV1 depletion reduced the chromatin-bound fraction of POLD1 and POLD3, while REV3 depletion (detected by RT-qPCR) reduced the chromatin-bound fraction of REV1, POLD1 and POLD3. As expected, POLD3 depletion did not affect the chromatin-bound fraction of REV1. Interestingly, binding of REV7 to chromatin was not affected by the depletion of REV1, REV3, or POLD3 (Fig. 5a–e). To substantiate the proposal that REV1/REV3 acts upstream of POLD1, we also analyzed whether depletion of REV1 could affect the localization of Pol δ in mitotic cells using immunofluorescence (Supplementary Fig. 8a). We observed that, upon depletion of REV1, POLD1 foci in mitotic cells were significantly reduced when cells were treated with APH for 16 h in S-phase (Supplementary Fig. 8b, d). Taken together, these data are consistent with the proposal that REV1/REV3 functions before Pol δ in the MiDAS pathway.

It was reported previously that REV1 regulates Pol ζ-dependent TLS in human cells through its interaction with REV7[68–70]. It is therefore intriguing that chromatin bound REV7 in prometaphase cells was not affected by depletion of REV1, REV3, or POLD3 following APH treatment. To address whether REV7 is essential for MiDAS, we analyzed MiDAS in U2OS cells in which REV7 was depleted using siRNAs (Supplementary Fig. 9a–d), or in a pair of isogenic U2OS cells with or without a functional *REV7* gene[71] (Supplementary Fig. 9e–h). We observed that loss of REV7 does not affect MiDAS. These data suggest that, during MiDAS, REV1 regulates Pol ζ through an interaction with proteins other than REV7. Hence, we sought to identify an alternative REV1 interactor that might be necessary for MiDAS. Based on the previous findings that REV1 can interact with other translesion polymerases containing a REV1-interaction region (RIR) motif[72], and the fact that POLD3 contains a RIR motif that has been predicted to direct binding to REV1[52], we tested whether the RIR motif in POLD3 might be essential for MiDAS. To this end, we created U2OS cell clones with doxycycline-inducible expression of the POLD3 cDNA that has the key $F_{238}F_{239}$ residues of the RIR motif substituted by $A_{238}A_{239}$ (Supplementary Fig. 10). Upon depletion of endogenous POLD3 using an siRNA targeting POLD3 mRNA 3′ UTR, the RIR-mutant POLD3 was not recruited to chromatin in early mitosis following APH treatment. Moreover, the frequency of MiDAS was greatly reduced in cells expressing this RIR-mutant POLD3 (Fig. 5f–h). Collectively, these data demonstrate that, in prometaphase, REV1 recruits POLD3 to perform MiDAS via an interaction with the RIR motif of POLD3.

**Loss of TLS renders cells vulnerable to oncogene activation induced RS**

Thus far, our findings indicate that the TLS polymerases REV1 and REV3, and the replicative polymerase Pol δ, act sequentially to promote MiDAS when cells are challenged with APH. Considering that oncogene activation is also able to induce RS in cancer cells, and the fact that some of the TLS polymerases are non-essential e.g. REV1[73], we investigated whether depletion of REV1 might specifically compromise the viability of cells in which oncogenes are activated. For this, we employed an established human cell model for oncogene activation, which is a U2OS cell line that overexpresses Cyclin E upon the

withdrawal of doxycycline from culture media[74] (Fig. 6a–c). As predicted, we observed that Cyclin E overexpression induced a significant increase of MiDAS (Fig. 6d, e). Moreover, in control cells lacking oncogene overexpression, we observed that the depletion of REV3, but not REV1, compromised cell survival (Fig. 6f–j). However, following the induction of Cyclin E expression, REV1 depletion also reduced cell survival (Fig. 6f–j). These findings are in agreement with the fact that REV1 is a non-essential polymerase, U2OS cells are viable in the absence of TLS[75], and that MiDAS is a salvage pathway to rescue UDRs, particularly in cancer cells[30,76–79]. Consistent with this effect being associated with MiDAS, we demonstrated that, when Cyclin E is overexpressed, the frequency of MiDAS is reduced significant upon depletion of either REV3 or REV1 (Supplementary Fig. 11). Taken together, our data suggested that REV1 protects cancer cells against the cytotoxic effects of oncogene activation.

## Discussion

MiDAS is a process that occurs in cells undergoing excessive replication stress. Two main models have been proposed recently to explain the mechanism by which MiDAS is conducted; namely, the BIR model discussed above and the 'symmetric fork cleavage' (SFC) model[80,81]. Both of these models posit that MiDAS depends on the cleavage of a stalled replication fork (Supplementary Fig. 12). Nonetheless, it was also reported that some MiDAS events depend upon RAD51[38,82,83] and may not depend on fork cleavage, but instead require de novo recruitment of RAD51 in mitosis[83]. Furthermore, it was recently suggested that MiDAS could be a continuation of DNA synthesis started in S phase and dependent on both RAD51 and RAD52[84]. Together, these findings have added another element of complexity to the issue of how MiDAS occurs. Without a doubt, it is reasonable to conclude that MiDAS is a phenomenon arising from several different pathways at various loci, which is perhaps not surprising considering that the underlying causes to replication perturbation at different loci in the human genome might vary. This variability is supported by the observation that MiDAS foci (EdU foci) on metaphase chromosomes can occur on only one of the two sister-chromatids, or alternatively on both of them[30,38,40].

In this study, we aimed to define the DNA polymerases required for MiDAS. Our data revealed that the two of the TLS enzymes, REV1 and REV3, as well as the replicative polymerase, Pol δ, are crucial for this process. Moreover, our results suggest that REV1/REV3 act upstream of POLD1, and that REV1 promotes MiDAS through its interaction with POLD3 via an RIR motif, while REV7 is not required in this process. Based on the BIR and SFC models for MiDAS discussed above, we propose that the utilization of both TLS and Pol δ are compatible with either of these models (Supplementary Fig. 12). Our proposal is that, following APH treatment in interphase, regions that are difficult-to-replicate promote RAD18-mediated PCNA monoubiquitylation, leading to the recruitment of REV1 and REV3 to the sites of the stalled DNA replication forks. In the BIR model, two potential roles of REV1/REV3 and Pol δ can be envisaged. First, the nascent D-loop might not be sufficiently stable to initiate conventional Pol δ-mediated DNA synthesis, because DNA end-resection is often very limited during the initiation of BIR[85]. Instead, this might require REV1/

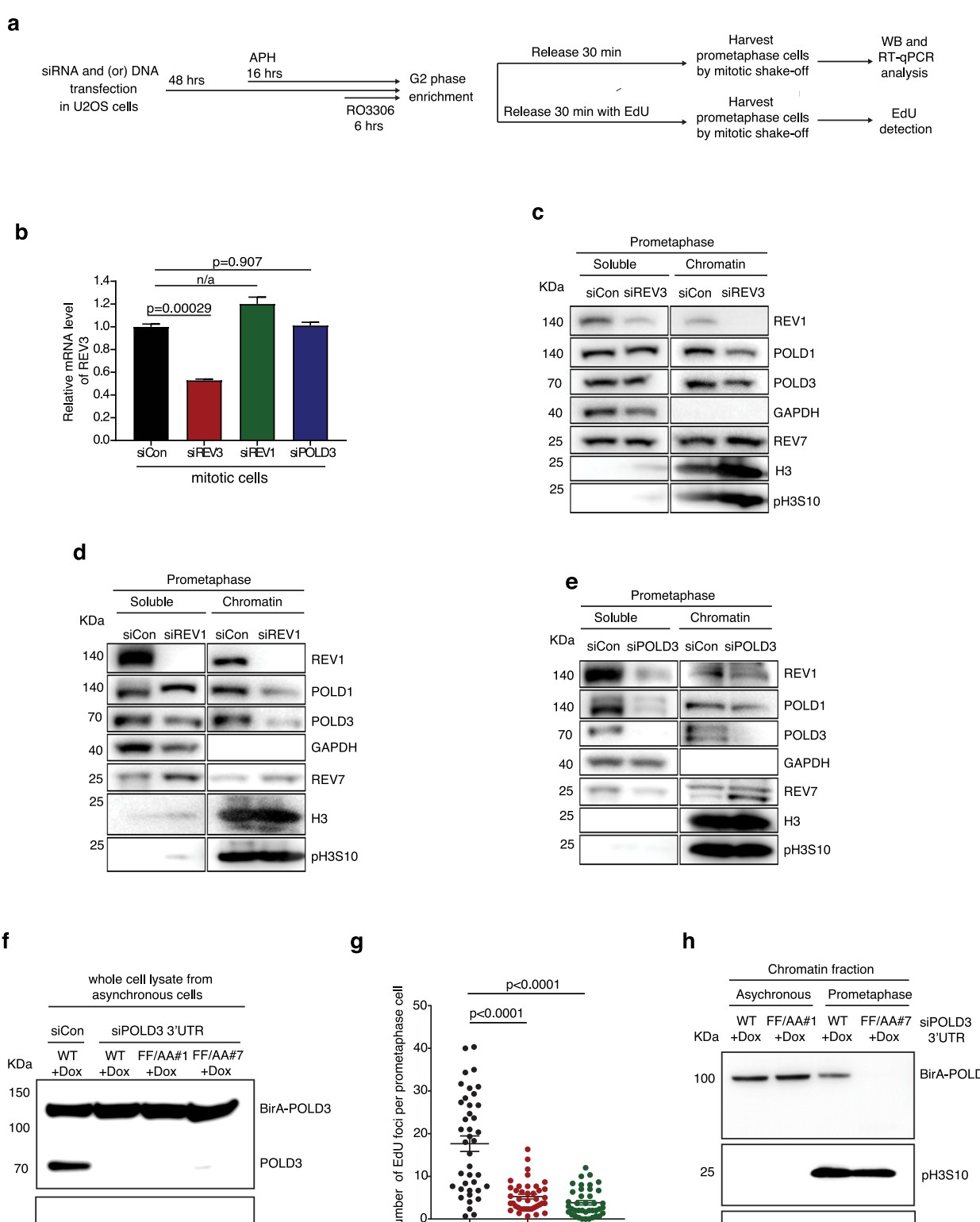

REV3 to initiate synthesis from the unstably paired invading strand. After the D-loop-based end extension is completed, POLD1 would then be recruited to the nascent leading strand via a POLD3-mediated polymerase switch, to allow rapid and processive DNA synthesis to be conducted. Second, it is known that BIR in yeast can involve the maturation of the migrating D-loop into a replication fork, which could be the point at which the polymerase switch would occur. In the alternative SFC model, REV1/REV3 would localize to stalled replication forks in S/G2 phase. Upon entering mitosis, the two leading strand templates would be cleaved by an SLX4-associated SSE. While the two broken ends could be ligated by end joining, the intact strand would require the gap filling initiation and extension ability of REV1 and REV3, followed by the POLD3-mediated polymerase switch to allow processive DNA synthesis if the gap to be filled was large. Considering a recent

**Fig. 5 | REV1 promotes MiDAS through its interaction with POLD3, but not with REV7. a** Experimental workflow for the analysis of chromatin-bound proteins in U2OS cells following REV1, REV3 or POLD3 depletion and APH treatment (panels **b**–**e**). For panels **f** to **h**, when studying the function of the POLD3 RIF motif in MiDAS, cells were transfected with siRNA targeting POLD3-3′UTR region or a control siRNA, in wild type or $F_{238}F_{239}/A_{238}A_{239}$ BirA-POLD3 mutant U2OS cells (WT, FF/AA#1, or FF/AA#2) respectively. **b** Quantification of REV3 mRNA levels by RT-qPCR in mitotic cells. The RT-qPCR value was normalized against a region of the GAPDH gene for each sample. Data of each bar are means of independent experiments. Error bars represent SEM. *P* values were calculated using a two-tailed Student's *t*-test (*n* = 3 biological replicates). n/a, not applicable. **c**–**e** Representative images of Western blot (WB) analysis of soluble or chromatin bound proteins from prometaphase U2OS cells using antibodies specific for REV1, POLD1, POLD3, or REV7 following depletion of REV3, REV1, or POLD3 respectively. GAPDH was used as a loading control for soluble fraction, Phospho-Histone H3 (pH3S10) was used as

a loading control for a mitotic chromatin-associated proteins, and Histone H3 (H3) as a loading control for chromatin-associated proteins. **f** WB analysis of exogenous BirA-POLD3 and endogenous POLD3 after transfecting cells with control or POLD3 3′UTR siRNAs in BirA-POLD3 WT or mutant U2OS cells. GAPDH was used as a loading control. **g** Quantification of MiDAS foci in prometaphase cells expressing wild type or $F_{238}F_{239}/A_{238}A_{239}$ BirA-POLD3 mutants following POLD3 depletion. Each data point in chart **g** is means of three independent experiments and plotted with Prism (*n* = number of cells analyzed in each condition in three experiments). Error bars represent SEM. *P* values were calculated using a two-tailed non-parametric Mann–Whitney test. **h** WB analysis of chromatin bound BirA-POLD3 in asynchronous or prometaphase BirA-POLD3 WT or mutant U2OS cells following endogenous POLD3 depletion by 3′ UTR siRNA. Phospho-Histone H3 (PH3S10) was used as a mitotic chromatin-associated loading control, and Histone H3 as a general chromatin-associated loading control. Hr hour, min minute.

report showing that yeast Rad52 cooperates with TLS polymerases when cells are challenged by MMC or UV induced damage[86], it is possible that RAD52 might also facilitate this process.

TLS polymerases exist in organisms ranging from bacteria to yeast and mammalian cells. They play a crucial role in bypassing small DNA lesions during DNA replication, which can lead to local mutations, but could limit extreme genetic instability caused by more extensive damage to DNA replication forks or persistent fork stalling. In human cells, the key TLS polymerases include Pol κ (POLK), Pol η (POLH), Pol ι (POLI), and REV1, all of which belong to the Y family of DNA polymerases, and Pol ζ, which belongs to the B family. While Pol κ, Pol η, Pol ι are mainly known to be recruited to MMC- or UV-induced stalled replicative DNA lesions and contribute to replicative gap filling[87], Pol ζ can add nucleotides to misaligned primers in collaboration with REV1[88]. Different from the conventional view of TLS polymerases acting over very short distances, a recent study indicated that yeast Pol ζ can conduct DNA synthesis over a distance of >200 bp downstream of UV-induced lesion[89]. In our study, we defined additional information concerning the DNA synthesis phase of MiDAS. These include, i) REV1 is recruited to UDRs that persist into early mitosis by RAD18-mediated PCNA monoubiquitylation; ii) REV1 acts in conjunction with REV3 to initiate DNA synthesis during MiDAS, and iii) REV1 then recruits POLD1 via the POLD3 RIR motif to conduct further DNA synthesis during MiDAS. Notably, neither Pol η nor REV7 is required for MiDAS. In the future, it would be interesting to define whether other human TLS enzymes might contribute to MiDAS.

To rule out the possibility that the siRNA treatments utilized could on their own activate a DDR in mitotic cells, we assessed γH2AX foci in mitotic cells using QIBC technology supported by a ScanR system (Olympus), where DNA content and immunofluorescence staining of proteins of interest can be assessed simultaneously and automatically (Fig. 7). Our results indicate that, in cells not exposed to APH, most of the siRNA treatments alone do not generate a significant increase in the frequency of γH2AX foci; the exceptions being siRNAs targeting CDT2 or Pol η. Following APH treatment, combined with siRNA treatment, we observed a significantly increased frequency of γH2AX foci in all cases, although it is reassuring that the overall level of DNA damage was comparable amongst the different APH/siRNA treatments. Again, CDT2 was an exception, but this is not surprising considering that CDT2 is known to be required for PCNA-ubiquitination in normal proliferating (unstressed) cells, as discussed above.

Given that MiDAS is elevated in cells challenged with RS and is particularly prevalent in aneuploid cancer cells with oncogene activation[30], inhibition of the polymerases playing a specific role in MiDAS might provide an opportunity for therapeutic intervention. Considering that inhibition of Pol ζ, Pol δ or REV3 is toxic to normal cells, and perhaps unlikely to provide a suitable therapeutic index in vivo, inhibitors specifically targeting REV1 or its interaction with POLD3 might be a promising avenue to pursue for the development of

a new cancer therapy. Interestingly, recently, it was shown that a small-molecule inhibitor targeting the C-terminal domain of REV1 could synergize with DNA-replication-gap inducing cancer treatments[75]. In the future, it would be interesting to validate whether this inhibitor could particularly affect cancer cells with elevated levels of MiDAS.

## Methods

### Cell lines and cell culture

The osteosarcoma cell line U2OS, cervical carcinoma cell line HeLa, and colorectal carcinoma cell line HCT116 were obtained from the American Type Culture Collection (ATCC). The U2OS-Cyclin E (Tet-off) cells were a kind gift from Prof. Thanos D. Halazonetis (University of Geneva, Switzerland). The U2OS REV7-/- cell line and its parental cell line were a kind gift from Prof. Alan D. D'Andrea (Harvard Medical School, USA). The U2OS Flp-In T-Rex cell line, and U2OS cell lines stably expressing StrepHA-PCNA(WT) or StrepHA-PCNA(K164R) were kindly provided by Prof. Niels Mailand (University of Copenhagen). All cell lines were cultured in Dulbecco's modified Eagle's medium (DMEM; ThermoFisher Scientific) supplemented with 10% fetal bovine serum (FBS; Sigma Aldrich) and 1% Pen/Strep (ThermoFisher Scientific). The cells were maintained at 37 °C in a humidified atmosphere with 5% $CO_2$ and were subjected to monthly mycoplasma testing (MycoAlert; Lonza). Only mycoplasma-free cells were analyzed.

### Cell treatment

To introduce replication stress, cells were treated with 0.4 μM aphidicolin for the period indicated in the figures. To induce POLD1 degradation, a modified HCT116 cell line that has endogenous *POLD1* targeted by CRISPR-Cas9 and exogenous stable POLD1-OsTIR1(F74G) expression (the HCT116-POLD1-AID2 cell line) was treated with 1 μM 5-Ph-IAA (BioAcademia) for the periods indicated in the Fig. 1a–e.

To enrich mitotic cells, asynchronous cells were treated with CDK1 inhibitor RO-3306 (7 μM; APExBIO) for 6 h, or in the last 6 h of APH treatment where necessary. Cells were then rinsed with pre-warmed medium (37 °C) for three times (within 5 min) before being released into fresh cell culture (with EdU, or other relevant treatment) to allow them to progress into mitosis for the follow-up analyses.

For MiDAS analysis in prophase/prometaphase, EdU was added in the pre-warmed medium (37 °C) and incubated with cells for 30 min. Prometaphase cells were then collected by mitotic-shake off, re-seeded onto Poly-L-Lysine-coated slides (Sigma Aldrich) for IF analysis, or collected by centrifugation for Western blotting (WB) analysis. To obtain anaphase cells, prometaphase cells collected from mitotic-shake off were re-seeded onto pre-warmed, Poly-L-Lysine-coated slides (Sigma Aldrich), and were incubated for additional 20 min to allow them to progress into anaphase for anaphase bridges analysis. To obtain G1 phase cells for micronuclei and 53BP1 foci analysis, prometaphase cells collected from mitotic-shake off were re-seeded onto

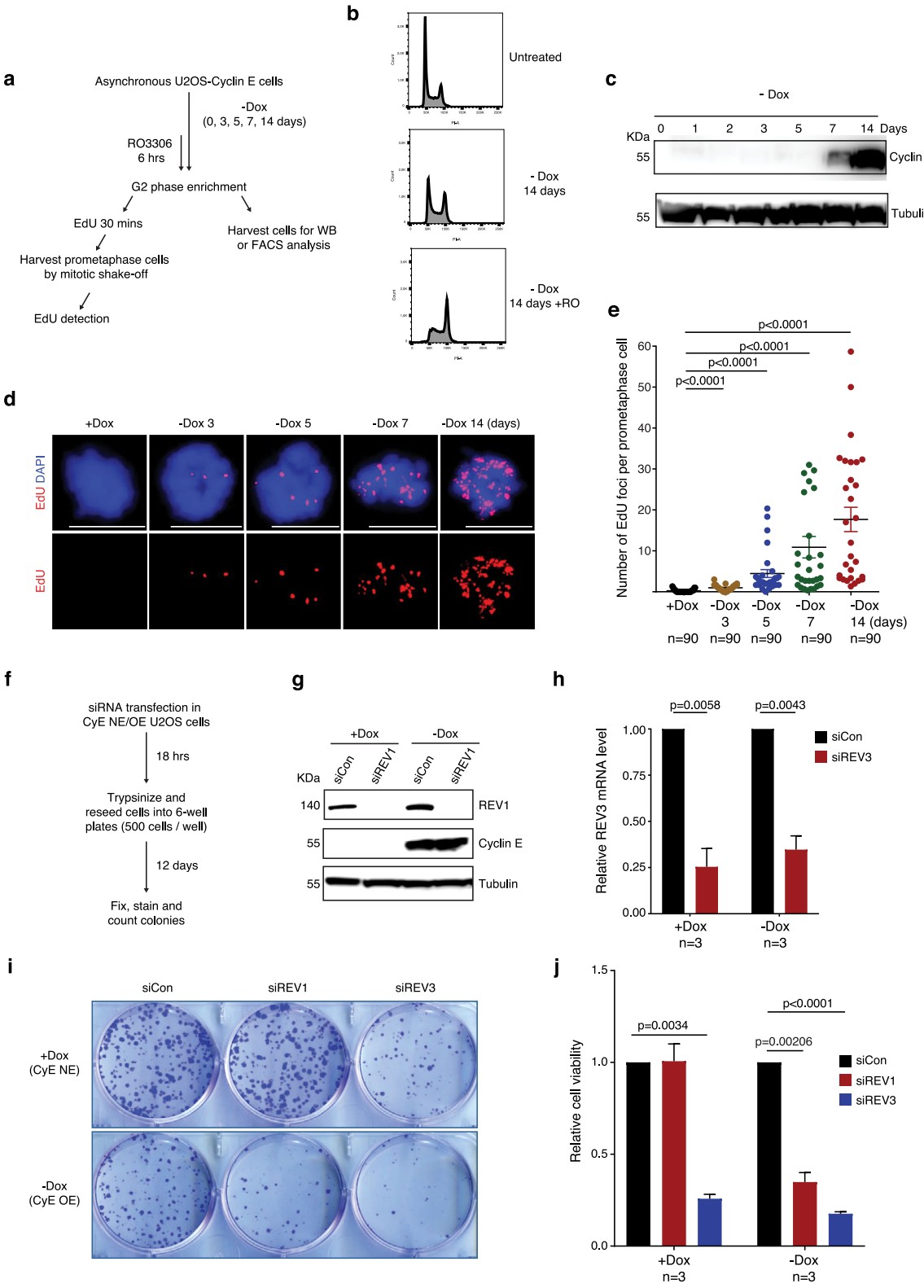

pre-warmed, Poly-L-Lysine-coated slides and were then incubated for an additional 2.5 h to allow them to progress into the next G1 phase.

## Plasmids
The POLD3 cDNA was synthesized by ThermoFisher Scientific and was used as the template to generate a POLD3 fragment with primers POLD3.F and POLD3.R primers. The pCDNA5_FRT-TO_HA-BirA*-POLD3

was constructed by inserting the POLD3 cDNA fragment into the pCDNA5_FRT-TO_HA-BirA* plasmid digested with KpnI and BamHI enzymes. To create pCDNA5_FRT-TO_ HA-BirA*-POLD3 ($A_{238}A_{239}$), mutations at $F_{238}F_{239}$ were introduced by 'QuikChange II XL Site-Directed Mutagenesis Kit' (Agilent technologies) using primers POLD3 (F238F239A238A239) F and POLD3 (F238F239A238A239) R. The REV3-3xFLAG (REV3FLAG) plasmid was a kind gift from Dr. C. E. Canman

**Fig. 6 | Depletion of REV1 renders cells sensitive to oncogene activation induced replication stress. a** Experimental workflow for cell treatment and analysis of MiDAS in U2OS-Cyclin E cells after removing doxycycline from culture medium at different time points to induce Cyclin E expression. **b** Representative flow cytometry histogram of propidium iodide (PI) fluorescence of U2OS-Cyclin E cells treated as shown in panel **a**. Untreated cells were used as a control. X-axis, total PI area; Y-axis, cell count. **c** Western blot (WB) analysis of Cyclin E expression after removing doxycycline from culture medium at different time points as shown in panel **a**. β-tubulin was used as a loading control. Representative images (**d**) and quantification (**e**) of MiDAS foci (labeled with EdU; red) in prometaphase cells treated as shown in panel **a**. DNA was stained with DAPI (blue). Scale bars, 10 μm. Each data point in chart **e** is means of three independent experiments and plotted with Prism ($n$ = number of cells analyzed in each condition in three experiments).

Error bars represent SEM. $P$ values were calculated using a two-tailed non-parametric Mann–Whitney test. **f** Experimental workflow for colony formation assays following REV1 or REV3 depletion in U2OS cells with no Cyclin E expression (CyE NE) or overexpression (CyE OE). **g** WB analysis of REV1 depletion after transfecting U2OS (CyE NE/CyE OE) cells with control or REV1 siRNA. β-tubulin was used as a loading control. **h** Quantification of REV3 mRNA by RT-qPCR after transfecting U2OS (CyE NE) and U2OS (CyE OE) cells with control siRNA or REV3 siRNA. The RT-qPCR value was normalized against a region of the GAPDH gene for each sample. Representative images (**i**) and quantification (**j**) of colonies formed following the treatment in panel **f**. Data in charts **h** and **j** are means of independent experiments. Error bars represent SEM. $P$ values were calculated using a two-tailed Student's $t$-test ($n$ = 3 biological replicates). Hr hour, min minute.

(University of Michigan Medical School)[56]. To analyze the contribution of REV3 in MiDAS, we created a mutant REV3-3xFLAG plasmid (REV3FLAGATA). In this construct, the REV3 catalytic 'YGDTDS' motif was changed to a catalytic dead 'YGATAS' motif using primers reported previously[57] and a 'QuikChange II XL Site-Directed Mutagenesis Kit'. The mutated constructs were verified by DNA sequencing. Oligonucleotides used for plasmids construction are listed in Key Resources table.

### Establishment of an HCT116-POLD1-AID2 cell line
A CRISPR-Cas9 plasmid to target the region adjacent to the stop codon of the *POLD1* gene (CTTCGGACCCCCTGGACCTG/agg) using pX330-U6-Chimeric_BB-CBh-hSpCas9 (Addgene #42230)[90] was constructed. A donor plasmid containing the mAID-Clover (mAC) tag with a hygromycin resistant marker was constructed following the published protocol[91]. Parental HCT116 cells expressing OsTIR1(F74G) were transfected with the CRISPR-Cas9 and donor plasmids followed by selection in the presence of 100 μg/mL Hygromycin B[54]. After colony isolation, clones harboring bi-allelic insertion at the target locus were verified by genomic PCR. The expression and degradation of POLD1-mAC was confirmed by WB against POLD1.

### Establishment of POLD3 (WT or $F_{238}F_{239}A_{238}A_{239}$) U2OS Flp-In T-Rex cells
The pCDNA5_FRT-TO_HA-BirA-POLD3 WT or mutant ($F_{238}F_{239}/A_{238}A_{239}$) construct were transfected into U2OS Flp-In T-Rex cells together with a pOG44 plasmid using FuGENE® HD Transfection Reagent (Promega), following the manufacturer's instructions. Stable cell clones were cultured under Hygromycin B selection (250 μg/mL). Doxycycline (1 μg/mL)-inducible cell lines expressing HA-BirA-POLD3 were confirmed by WB analysis.

### RNA interference (RNAi)
RNAi was performed using Lipofectamine RNAiMax reagent (ThermoFisher Scientific) according to the manufacturer's instructions. ON-TARGETplus Non-targeting Pool siRNA (Dharmacon) or other gene specific siRNAs used are shown in the Key Resources Table. In each case, 80 nM siRNAs were used to target expression of a specific protein. All of the siRNAs used in this study are listed in Supplementary Table 1.

### Immunofluorescence analysis
Following the different treatments described above, cells growing on coverslips were fixed with PTMEF fixation buffer (4% paraformaldehyde, 200 mM PIPES, pH 6.8, 200 mM $MgCl_2$, 10 mM EGTA, pH 8, 0.2% Triton X-100) for 10 min at room temperature (RT), and washed once with 1x PBS for 5 min. Cells were then blocked with blocking buffer (3% BSA in 1x PBS containing 0.5% Triton X-100) for at least 1 h at RT or stored at 4 °C overnight. After blocking, cells were incubated with primary antibody diluted in blocking buffer at 4 °C overnight followed by 3 washes with blocking buffer (20 min each time). Cells were then

incubated with secondary antibodies diluted in blocking buffer for 1 h at RT and washed 3 times with blocking buffer (20 min each time). Slides were then mounted with DAPI-containing Vectashield medium (Vector Laboratories). Images were captured using an Olympus BX63 microscope and analyzed with CellSens (Olympus). All images were processed with Image J using the same settings. In the steps of counting foci, the samples were coded and counted manually. Antibodies used are listed in Supplementary Table 1.

### Ultrafine anaphase DNA bridge (UFB) detection
UFB detection was performed following a previously published protocol[92]. Briefly, anaphase cells were first fixed with PTMEF buffer for 10 min at RT and washed once with 1x PBS for 5 min. Cells were then blocked with blocking buffer overnight at 4 °C. After blocking, cells were incubated with primary antibody diluted in blocking buffer overnight at 4 °C followed by 3 washes with blocking buffer (20 min each), and then incubated with secondary antibodies diluted in blocking buffer at RT for 1 h. After three additional washes with blocking buffer, slides were mounted with DAPI-containing Vectashield medium (Vector Laboratories) before imaging. In the steps of counting UFBs, the samples were coded and counted manually.

### EdU detection
To detect EdU, cells were first blocked with blocking buffer (3% BSA in 1x PBS containing 0.5% Triton X-100) for 30 min at RT. EdU detection was then performed using Click-IT chemistry following the manufacturer's instructions (Click-IT EdU; Alexa fluor 594, Imaging Kits, Life Technologies). Images with EdU foci were captured using an Olympus BX63 microscope and analyzed with CellSens (Olympus). Images were then processed with Image J using same settings, and foci were counted manually with samples identity unknown to the counter.

### Western blot analysis
To obtain whole cell lysates, cells were harvested and lysed in cell extraction buffer (ThermoFisher Scientific) supplemented with a protease inhibitor cocktail and phosphatase inhibitors (Roche) for at least 30 min at 4 °C. The lysate was then subject to sonication (2 times; 10 s ON, 10 s OFF with maximum Power) using an MSE Soniprep 150 Plus sonicator. To obtain chromatin bound proteins, cells were first incubated with 1× RIPA buffer supplemented with protease inhibitor cocktail and phosphatase inhibitor for 5 min at 4 °C, and then the nuclei were harvested by centrifugation at 10,000 $g$ for 10 min at 4 °C. Subsequently, the nuclei were reconstituted using cell extraction buffer supplemented with protease inhibitor cocktail, phosphatase inhibitors and 0.1% (v/v) Benzonase® Nuclease (Sigma Aldrich) for 1 h at 4 °C, followed by sonication at 4 °C using a water bath sonicator (The Bioruptor® Pico; Diagenode, B01060010) (setting: 30 s on / 30 s off, 10 cycles for two rounds). Protein concentration was determined using a BCA protein assay (Pierce™ BCA Protein Assay Kit; Thermo Scientific). Aliquots of samples containing NUPAGE SDS sample buffer (ThermoFisher Scientific) were heated for 15 min at 70 °C and were

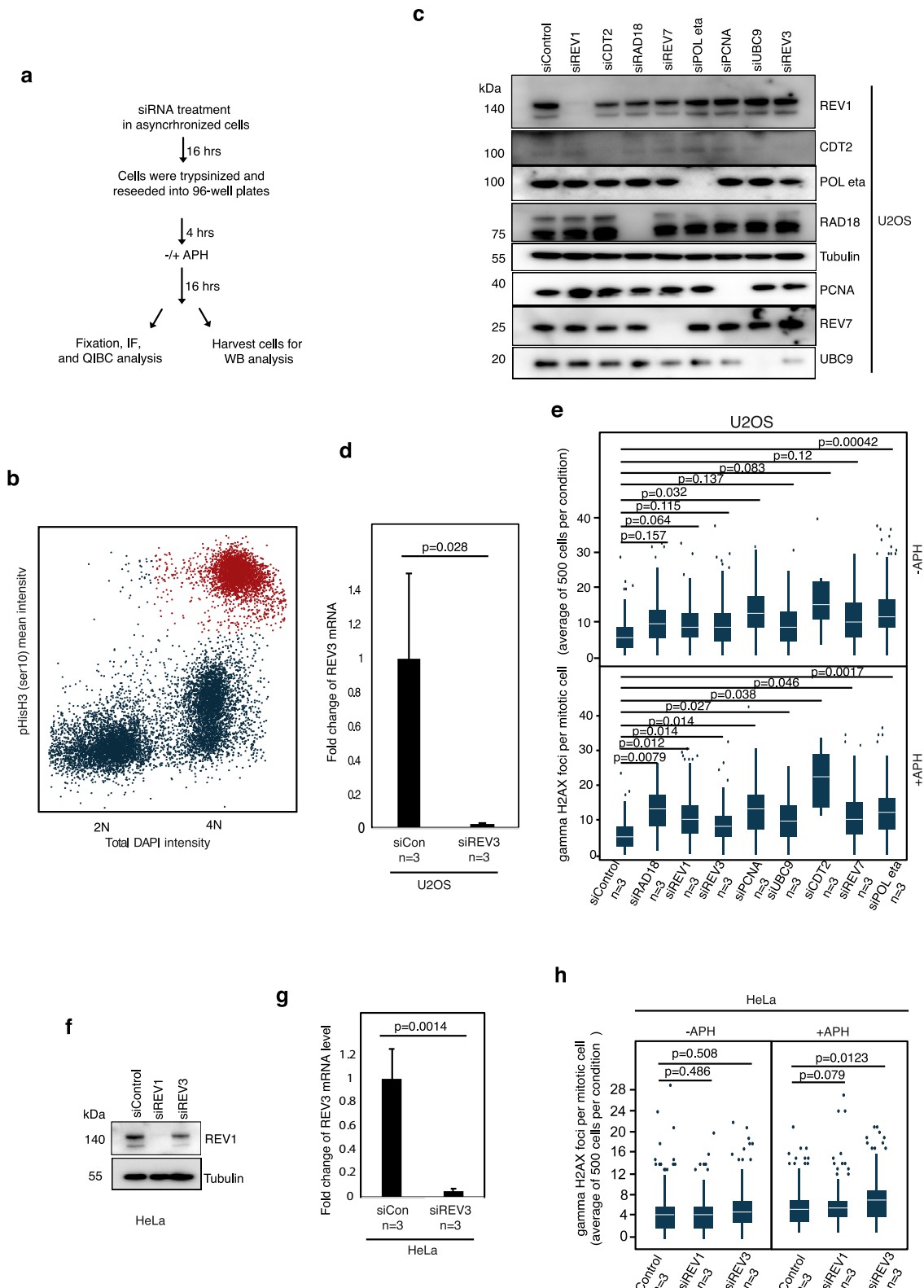

loaded into a NuPAGE 4–12% Bis-Tris Protein Gel (ThermoFisher Scientific). For protein obtained from whole cell lysate, 40 µg of protein was loaded for each sample. For protein obtained from chromatin bound fraction, 20 µg of protein was loaded for each sample. After SDS–PAGE, proteins were transferred onto a Hybond-PVDF membrane (Amersham Pharmacia) and were probed with specific antibodies according to the following protocol. The membrane was first blocked with 5% non-fat milk (Sigma Aldrich) in PBST (1x PBS containing 0.2% Tween-20) buffer for 1 h at RT, and then was incubated overnight with primary antibody diluted in blocking buffer at 4 °C. After 3 washes in PBST (20 min each), the membrane was incubated with secondary antibody diluted in blocking buffer for 1 h at RT, followed by 3 washes in PBST (20 min each). The WB signal was detected using the Luminata Forte HRP substrate (Millipore), and images were captured using an

**Fig. 7 | Assessment of replication stress during mitosis. a** Experimental workflow for assessment of replication stress level (evaluated by gamma H2AX foci; γH2AX) following APH and different siRNA treatments in U2OS or HeLa mitotic cells using QIBC analysis (Quantitative image-based cytometry). **b** Representative scatter plot generated by QIBC showing the cell cycle distribution of U2OS cells based on DNA content (total intensity, DAPI; blue) and level of Histone H3 phosphorylation (Ser10) (mean intensity; red). Each dot represents a single cell. Cells with 4 N DNA content and the highest level of Histone H3 phosphorylation (Ser10) were considered as mitotic cells (red) and subjected to further gamma H2AX foci quantification. **c** Representative western blot (WB) analysis of relevant proteins upon siRNA treatment as indicated. **d** Quantification of REV3 mRNA by RT-qPCR in U2OS cells. **e** Box plots of quantification of gamma H2AX foci per nucleus of mitotic U2OS cells by QIBC analysis. **f** WB analysis of relevant proteins following siRNA treatments in HeLa cells. **g** Quantification of REV3 mRNA by RT-qPCR in Hela cells. In charts **d** and **g**, error bars represent standard deviation (SD) and p values were calculated using a two-tailed Student's t-test (n = three independent experiments). **h** Box plots of quantification of gamma H2AX foci per nucleus of mitotic HeLa cells by QIBC analysis. In box plots **e** and **h**, the white horizontal lines represent the average number of gamma H2AX foci per mitotic cell; the box boundaries represents upper (75th percentile) and and lower quartile (25th percentile); the whiskers mark the range of the vertical scale from the highest (95th percentile) or lowest (5th percentile) of the displayed reference points; dots above the vertical line represent outliers; p values were calculated using a two-tailed Student's t-test (n = 3 independent experiments). In each condition, 500 cells were scored for gamma H2AX foci.

Imager 600 (Amersham). Quantifications for WB were performed using Image J/FIJI. Antibodies used are listed in Supplementary Table 1.

### Quantitative reverse transcription PCR (RT-qPCR)
RNA was extracted using an RNeasy Plus Mini Kit (Qiagen) following manufacturer's instructions. cDNA was generated using a reverse transcription kit (ThermoFisher Scientific) according to the manufacturer's protocol. qPCRs were performed with *Power*SYBR Green PCR Master Mix (Applied Biosystems) on a StepOnePlus Real-Time PCR machine (Applied Biosystems), and were analyzed with StepOne software v2.3 (ThermoFisher Scientific). The thermo-cycles for the qPCRs were: 1 cycle of 95 °C for 10 min, followed by 40 cycles of denaturation (95 °C for 15 s), annealing and extension (60 °C for 1 min). Primer sequences are listed in Supplementary Table 1. The delta-delta Ct method ($2^{-\Delta\Delta Ct}$) was used to calculate the relative fold gene expression of samples using GAPDH as a control.

### Colony formation assays
To test how oncogene activation affect cell survival in combination with either REV1 or REV3 depletion, the U2OS-Cyclin E (Tet-off) cells were cultured with cell culture with no doxycline for 14 days, and then re-seeded for siRNA treatment. 24 h after siRNA transfection, cells were trypsinized, counted and re-seeded in duplicates into 6-well plates (500 cells/well). Cells were cultured for additional 7–12 days to allow visible colony formation. Colonies were washed once with 1x PBS and were then fixed and stained using crystal violet solution (5% solution of crystal violet, 20% ethanol) for 45 min and then washed until the background staining was removed. Once the colonies were dried, they were counted manually with sample identity hidden from the counter. Each experiment was performed three times. Relative cell survival or colony formation was calculated by dividing colony counts from REV1 or REV3 depleted samples by the siRNA control treated samples (siCon).

### Quantitative image-based cytometry (QIBC) analysis of γH2AX foci in mitotic cells
Cells were trypsinized and re-seeded into in 96-well plates (Greiner; Cat no 650185) (15,000 cells per well). After the treatments indicated in Fig. 7, cells were pre-extracted with 0.4% Triton-X in PBS on ice for 4 min followed by fixation with 4% PFA in PBS (RT, 10 min). Fixed cells were then washed with 1x PBS for three times and were blocked with 3% BSA in 1x PBS and 0.4% TritonX (PBSAT). Finally blocked cells were incubated with antibodies against pS10Histone H3 (S10) (1:2000) and γH2AX (1:2000) in PBSAT buffer for overnight in 4 °C. The plates were then washed with PBSAT buffer 4 times, 15 min each at RT, and then incubated with Alexa Fluor488 (1:2000) or 568 conjugated (1:2000) secondary antibodies specific to mouse or rabbit IgG, respectively. Following the same washing steps as for the primary antibodies, DAPI was added to each well and the cells were imaged (typically 500 cells per condition) using a motorized Olympus IX-83 wide-field microscope in a ScanR system (Olympus) under non-saturating conditions.

The imaging platform was equipped with a Spectra X-LIGHT engine Illumination system with 6 color LEDs, DAPI, FITC, Cy3, and Cy5 fluorescent dyes compatible filter cubes, emission filters, and a Hamamatsu Camera Orca Flash 4.0 V2. An Olympus Universal Plan Super Apo 40x Objective (NA 0.9) was used for all QIBC data. Images were acquired and processed using the ScanR image analysis software, where a dynamic background correction and z-projection was applied. Segmentation of nuclei was performed using the DAPI signal with an integrated intensity-based object detection module, and segmentation of foci was performed with an integrated spot-detection module. Image capture and analysis settings were kept identical in all samples within each experiment. TIBCO Spotfire software was used to plot the total intensities for DAPI along the X-axis using arbitrary units (A.U.) and mean intensities (total intensities divided by nuclear area) for pS10Histone H3 (S10) along the Y-axis in color-coded scatter diagrams in a flow-cytometry-like fashion. Box plots were prepared using TIBCO Spotfire software representing the number of average γH2AX foci per nucleus under each condition. Within each experiment, similar cell numbers were compared for the different conditions.

### Flow cytometry
Following various treatments, cells were harvested and then fixed by drop-wise addition of ice-cold 70% ethanol to the cell pellet with gentle mixing of cells and the ethanol during the process. The cells were then incubated at −20 °C for 16 h. Fixed cells were centrifuged at 2000 g for 20 min at 4 °C and the cell pellet was washed two times with 1× PBS (500 g x 5 min). Following that, the cells were stained for 30 min at 37 °C with 80 μg/ml propidium iodide (Invitrogen) and 100 μg/ml RNase A (Sigma Aldrich) dissolved in 1X PBS. Fluorescence-activated cell sorting (FACS) analysis for each sample was carried out on a FACSCelesta flow cytometer (BD Biosciences). 50,000 cells were analyzed for each condition, and data was extracted using BD FACSDiva software, and displayed with FlowJO software.

### Statistics and reproducibility
Data from three independent experiments were presented as mean ± SEM, except in Fig. 7d, g, where data from three independent experiments were presented as mean ± SD. A non-parametric Mann–Whitney U test or two-tailed Student's t-test was used for statistical analysis, as indicated in each figure legend.

### Reporting summary
Further information on research design is available in the Nature Portfolio Reporting Summary linked to this article.

## Data availability
The data supporting the findings of this study are available from the corresponding authors upon reasonable request. The source data for graphs/statistics, western blots are available with in the Source Data

file. The source data for all of the images in figures are deposited at Figshare via https://figshare.com/s/e17deec87055270ae64d. Source data are provided with this paper.

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

## Acknowledgements

We thank Ian D. Hickson for inspiring discussions and critical reading of the manuscript. Wei Wu was supported by the Chinese National Natural Science Foundation (82103232; W.W.). The work in the Liu laboratory is supported by the European Union (H2020/Marie Skłodowska-Curie Actions; 859853; Y.L.), Danish Independent Research Fund (1030-00180B; Y. L.), and Danish National Research Foundation (DNRF115; Y.L.). The work in the Kanemaki laboratory is supported by the JSPS KAKENHI grants (20H05396 and 21H04719; M.T.K.).

## Author contributions

W.W. and Y.L. conceived the study, designed experiments, and contributed to data analysis. W.W., S.B., R.B., and M.T.K. contributed to the design of the experiments, performed the majority of them, and analyzed the data. K.L. performed chromatin binding and colony formation assays, M.M.G.D. performed cell cycle analysis, and M.C. contributed to colony formation assays. W.W. and Y.L. wrote the manuscript and all authors edited it. Y.L. supervised the study.

## Competing interests

The authors declare no competing interests.
