## [Peer Review File · Nature Communications]

Mitotic DNA synthesis in response to replication stress requires the sequential action of DNA polymerases zeta and delta in human cellsREVIEWER COMMENTS

Reviewer #1 (Remarks to the Author):

In this manuscript the authors identify factors required for MIDAS. MIDAS (mitotic DNA synthesis) is a recently described repair pathway to complete replication of genomic regions that remained unreplicated during interphase. The genomic regions affected by MIDAS include the common fragile sites (CFSs) that are frequently targeted in human cancers. As such, understanding MIDAS at a mechanistic level is of great interest. The authors present new data implicating PCNA ubiquitination, REV1 and the rev1-interacting region of POLD3 in MIDAS. These results are interesting, compelling and worthy of publication in Nature Communications. The authors also implicate pol-zeta in MIDAS. These results need to be better documented. It is possible that further experiments may show that pol-zeta is not implicated in MIDAS. This would be fine, as the manuscript still makes important contributions to the field and is worthy of publication.

Specific points:

1. Figs 1A-G show the establishment of cells, in which POLD1, the catalytic subunit of pol-delta, can be inducibly degraded and that degradation of POLD1 suppresses MIDAS. Figs 1H-K show that depletion by siRNA of REV3 (the catalytic subunit of pol-zeta) or REV1 (which interacts with POLD3, a subunit of the four-subunit pol-zeta polymerase: Rev3–Rev7–PolD2–PolD3) also suppresses MIDAS. These results implicate pol-delta and pol-zeta in MIDAS. A role of pol-delta in MIDAS was anticipated, whereas a role of pol-zeta was unexpected.
2. Fig. 2 shows that PCNA ubiquitination at K164 is required for MIDAS and for recruitment of Rev1 to chromatin in mitotic cells. These results suggest that PCNA K164 ubiquitination in aphidicolin-treated cells, results in the recruitment of Rev1, which, as shown in Fig. 1, is needed for MIDAS. In contrast, depletion of UBC9 does not affect MIDAS, which suggests that PCNA sumoylation is not needed for MIDAS.
3. Fig. 3 explores which is the ubiquitin ligase responsible for PCNA ubiquitination. Depletion of RAD18 suppresses aphidicolin-induced PCNA ubiquitination, whereas CDT2 depletion enhances aphidicolin-induced PCNA ubiquitination. Depletion of RAD18, but not of CDT2, also suppresses MIDAS. Thus, RAD18 is responsible for PCNA ubiquitination and MIDAS.
4. Fig. 4 examines at what stage POLD1 and REV3 function in MIDAS. The authors claim that ultrafine anaphase bridges that stain with RPA represent advanced homologous recombination intermediates, whereas ultrafine anaphase bridges that stain with PICH represent early homologous recombination intermediates. Wild-type aphidicolin-treated cells have very few ultrafine anaphase bridges. Depletion of POLD1 leads to anaphase bridges that stain with RPA, whereas depletion of REV3 to anaphase bridges that stain with PICH. The effect of POLD1 depletion is quite strong, whereas the effect of REV3 depletion is not that strong. These results are interesting and suggest that POLD1 may have a greater role in

MIDAS than REV3.

5. Figs 5A-C show that when REV1 is depleted, then POLD1 and POLD3 are not recruited to the chromatin fraction in prometaphase cells. For reasons related to the next point, it would have been interesting to show that REV3 is also not recruited to the chromatin fraction and to also explore whether REV7 is recruited to the chromatin fraction.

6. Figs 5D-F show that depletion of REV7 or a compound that inhibits the interaction between REV1 and REV7 did not affect MIDAS. This observation is surprising, because the pol-zeta four-subunit polymerase consists of REV3, REV7, POLD2 and POLD3. Thus, the question arises whether the role of POLD3 in MIDAS implicates pol-zeta or pol-delta or both in MIDAS. If only pol-delta is implicated, then the findings would still be of interest, since this study clearly implicates PCNA ubiquitination, REV1 and the interaction between REV1 and POLD3 (Figs 5G-I) in MIDAS. I wonder, if the authors can resolve this issue in a revised version of the manuscript, for example by addressing the question in point 5, above, or by any other experiments they can think of (for example, using more siRNAs targeting REV3 to validate the role of REV3 in MIDAS, etc).

7. Figs 6A-D show that cyclin E overexpression induces MIDAS. This is a very nice and interesting result.

8. Figs 6E-I show that depleting REV1 suppresses growth (colony formation) of cyclin E overexpressing cells, but not of cells expressing normal levels of cyclin E. In contrast, REV3 depletion affects colony formation of both cyclin E overexpressing cells and cells expressing normal levels of cyclin E. It would be nice to show that MIDAS in cells overexpressing cyclin E requires REV1 and/or REV3.

9. Since REV3 depletion suppresses colony formation also of cells with normal levels of cyclin E (which cells do not have MIDAS), these results again raise some questions about whether pol-zeta (REV3) has a role in MIDAS. Irrespective of whether pol-zeta has a role in MIDAS, the effect of REV1 specifically on cells overexpressing cyclin E is interesting and worth reporting.

Reviewer #2 (Remarks to the Author):

Wu et al., NCOMMS-21-35946-T

In this manuscript, Wu et al explore a role for the translesion synthesis (TLS) DNA polymerases REV1 and Rev3 in MiDAS, the replication during mitosis of fragile sites, induced by low concentrations of Aphidicolin.

Briefly, the authors do convincingly show that these TLS polymerases operate during MiDAS, dependent on PCNA ubiquitination by RAD18 and on FANCD2. Furthermore, the authors provide evidence that this process requires the interaction between REV1 and the RIR motif of POLD1, a subunit of replicative DNA polymerase delta that is also part of the REV1/REV3 complex. Finally, the authors show that inhibition of

REV1 sensitizes cells to oncogene-induced replication stress.

I have mixed feelings about this manuscripts. The experiments mostly appear well executed and interpreted (but see below) and most conclusions appear warranted. However, I do have significant problems with the (lack) of novelty of the work. For example, there is an extensive body of literature on the involvement of RAD18-mediated PCNA-Ubiquitination and FANCD2 in REV1/REV3-mediated TLS but this (virtually) is not cited. E.g., a Medline search yielded 8 papers on the involvement of FANCI/FANCD2 but none of these are cited. Also, since the factors involved in MiDAS by REV1/REV3 are essentially the same as for 'normal' REV1/REV3-dependent TLS, that also operates beyond S phase (PMID: 21908406, not cited), one wonders whether MiDAS by these TLS proteins does not represent a delayed form of 'normal' TLS. It would be interesting if this is the case by investigating whether there are differential requirements for TLS and MiDAS by REV1/REV3. Moreover, MiDAS has been described to require break-induced replication or symmetric fork cleavage and the authors assume but provide no evidence that REV1/REV3-dependent MiDAS is associated with either of these breaks repair pathways (that are inappropriately extensively are described in the Discussion). Then the sensitization of cells with oncogenic replication stress by REV1 inhibition also has been described before (PMID: 32577513, not cited here). In conclusion, I found the paper lacking in novelty and depth.

Specific comments on the text:

1. Line 39: TLS is no DNA repair
2. Line 92: 'the end G2 phase' should read 'the end of G2 phase'
3. Line 226: CDT2 depletion unexpectedly induces PCNA Ubiquitination and a consequent increase in MiDAS (or TLS?). This unexpected finding is not at all explained. Does this imply that CDT2 acts dominantly negative over RAD18?
4. Line 282: RIR-mutant POLD3 has less MiDAS (also less TLS during G2?). But this is no direct evidence that 'REV1 recruits Pol δ to perform MiDAS via an interaction with the RIR motif of POLD3

Comments on the experiments and Figures:

1. Graphical abstract: "Replicaltion" should read Replication.
2. Figure 1 D: Provide FACS plots to show that Ro3306 treatment for 6 hours results in accumulation of G2 cells.
3. Figure 2 A: It is not clear which antibodies were used for this Western blot.
4. Figure 2 G: Provide a control showing non-chromatin-bound REV1.
5. Figure 2 G: What is the extra band indicated with an asterisk? This band is absent in the lane labelled with siCon.
6. Figure 3 B: What is the fate of REV1 when cells are treated with siCDT2? Is there more chromatin-bound Rev1?
7. Figure 3 E: What is the effect of RAD18 depletion on RPA-coated/PICH-coated UFBs in anaphase?
8. Figure 4 A: Why did the authors treat the cells with 5-Ph-IAA for only 30 minutes, knowing that POLD1 is still detectable under these conditions (Fig 1C)?
9. Figures 4 C + E: Y-axis labelling is not clear. The number of RPA- positive UFBs or PICH-positive UFBs per anaphase cell can only be shown as rounded numbers.

10. Figure 5 B: Why is there more POLD3 (and not POLD1) in asynchronous cells depleted from Rev1?
11. Figure 5 E: If REV1 recruits POLD1 to perform MiDAS, then treatment of cells with siRNA Rev1 should inhibit MiDAS. This is an obvious experiment that should be included in the paper. The experiment with the JH-RE-06 is not sufficient, since there is no positive control that this compound is able to inhibit REV1/REV7.
12. Figure 6 A: Provide FACS plots showing that cells are synchronized by the Cyclin E overexpression/Ro3306 treatment.

Reviewer #3 (Remarks to the Author):

This manuscript reports the mechanism underlying mitotic DNA synthesis (MiDAS), which takes place in cells exposed to mild replication stress (RS), as well as in cancer cells harbouring intrinsic RS. Despite its implications in cancer, the complete picture of MiDAS remains enigmatic.

It has previously been demonstrated that MiDAS is dependent on POLD3, which is involved in both Pol δ -dependent (replicative) and Pol ζ -dependent (translesion) DNA synthesis. In this study, the authors aim to determine how these polymerases act during MiDAS.

First, using an elegant AID-degron system that allows rapid depletion of POLD1, they confirmed that Pol δ indeed promotes MiDAS. They then used the siRNA-mediated depletion method to demonstrate that the REV1 and REV3 components of Pol ζ also support MiDAS. Pol η , on the other hand, appears not to be involved in MiDAS.

They also demonstrated that PCNA promotes MiDAS through the K164 residue, which is targeted for ubiquitylation by RAD18 (upon DNA damage) or CDT2 (during normal replication), or sumoylation by UBC9. Once again, using the siRNA approach, the authors demonstrated that RAD18, but not CDT2 nor UBC9, promotes MiDAS. All these observations align with the idea that stress-induced PCNA K164 mono-ubiquitylation recruits Pol ζ and potentially promotes translesion DNA synthesis in mitosis.

They then aimed to clarify how Pol ζ and Pol δ act during MiDAS. They found that POLD1 depletion increased RPA-positive UFB (considered to reflect MiDAS initiation), while REV3 depletion induced PICH-positive UFB (considered to reflect late replication intermediates, but not MiDAS initiation). Also, upon REV1 depletion, POLD3 and POLD1 recruitment to prometaphase chromatin was impaired. The depletion of REV7, which is proposed to promote Pol ζ -dependent DNA synthesis, had no impact on MiDAS. A POLD3 missense mutant, which is defective in REV1 interaction, however, failed MiDAS. Based on these observations, the authors propose that the MiDAS is initiated by PCNA K164 ubiquitylation, which in turn recruits REV3 (Pol ζ), REV1, POLD3 then POLD1 (Pol δ) in this order.

Finally, they show that U2OS cells overexpressing cyclin E, hence inducing oncogenic RS, were sensitized

by the depletion of REV1 or REV3. Accordingly, they propose that the MiDAS inhibition by blocking Pol ζ -dependent translesion DNA synthesis might be an effective way to treat cancer cells.

The manuscript is well written, largely logical, and the results look very nice too. It is worth noting, however, that MiDAS events in siRNA treated cells (i.e. long-term depletion of proteins of interest) require careful assessment of the level of overall RS and associated DNA damage, especially when down-regulating factors which play significant roles during interphase. Indeed, this reviewer assumes that the increased MiDAS in siCDT2 treated cells (Fig 3E, F) reflects an increase of RS upon the depletion of CTD2, which targets a variety of proteins involved in cell cycle progression. In this context, this reviewer recommends that the authors show the degree of overall DNA stress (e.g. γ H2AX foci) in mitotically collected cells in each siRNA treated condition, ensuring the comparative assessment of the level of MiDAS relative to the level of DNA stress in mitosis. It will also be good to confirm the successful depletion of target protein in mitotically collected cells, as is shown in Fig 5B and I for chromatin-associated REV1 and POLD3, respectively.

Determining the order of the Pol ζ and Pol δ action appears somewhat tricky. This reviewer found the interpretation of the results shown in Fig 4 highly speculative. Also, it is not appropriate to directly compare phenotypes of 5-Ph-IAA treated cells and siREV3 treated cells. Fig 5 is more straightforward to interpret, although it would be good to see the level of chromatin-associated POLD1/3 upon REV3 depletion, and conversely chromatin-associated REV3 upon POLD1/3 depletion. Similarly, since POLD1/3 foci can be detected by IF (as shown in Fig S2 in this study; Minocherhomji et al, 2015), the authors may want to assess the levels of RS-induced mitotic POLD1/3 foci upon REV1/3 depletion to consolidate their model more directly. The notion that Pol ζ catalytic activity may not be involved in MiDAS is intriguing. This can be tested directly by complementing REV3 depleted cells (which is defective in MiDAS) with a catalytically inactive REV3 missense mutant. This is feasible as they already have a FLAG-REV3 expression construct (Fig 1I)

The reduced cell survival of cyclin E overexpressing U2OS cells upon REV1/3 depletion is also interesting, but this reviewer considers that the REV1/3-mediated TLS in interphase can explain this phenotype.

The numbers of cells analyzed ($n > 90$ in total?) in each condition are somewhat modest for the study of this sort. In the method section & the reporting summary, it is stated that three biological replicates have been conducted, while analyses were not blinded. They should describe how foci were counted and justify their methodology.

Regarding the final model, this reviewer wonders if the data shown in this manuscript may explain MiDAS in the broader context beyond what the authors frame around the SSE-dependent BIR/SFC-mechanisms (Fig S7). Indeed, translesion DNA synthesis is best characterized as a mechanism that allows the DNA replication machinery to bypass DNA lesions. Hence, it seems reasonable to speculate that Pol ζ first bypasses DNA lesions generated at stalled forks, and then Pol δ takes over to carry out long-range DNA synthesis. In line with this notion, a MiDAS mechanism independent of mitotic DNA cleavage has been recently proposed (Wassing et al., Nat Comm, 2021).

Lines 123-129 (also lines 360-363)

The MiDAS model presented appears highly biased. For example, while TRAIP is shown to remove the replication machinery upon mitotic entry (mainly demonstrated in *Xenopus* and worm; and more recently in mouse ES cells, Villa et al. EMBO J, 2021), data presented in Minocherhomji et al (Fig 4a and b, Nature, 2015) indicate that many components of the replication machinery (incl. a known TRAIP substrate MCM7, as well as other MCMs, POLD1, 3 and PCNA) are retained or recruited on mitotic chromatin in cells exposed to RS in U2OS. Critically, TRAIP was identified as a PCNA binding protein (Hoffmann et al., 2015) and proposed to target PCNA itself (Feng et al. 2016). Given that the current study focuses on the role of the replication machinery and PCNA during MiDAS, a broader and critical consideration of the model in the context of different systems (experimental conditions, model organisms and cell lines) will be appropriate.

Minor points

Line 110

SLX4-deficient cells phenotype can be explained not solely by MiDAS event, as SLX4 can resolve HR intermediates generated in interphase.

Lines 133, 136 & 140

Introduce REV1, 3 and 7 properly.

Fig 2

What is the band appearing in siPCNA treated cells (marked by *) in Fig 2G? The positions of molecular weight markers for PCNA blot in Fig 2A, B and G look inconsistent.

In many Figure panels where experimental design is depicted (Figs 1D, 3D, 4A, 6A, S1D, S3A, S4A, S5D), the timing of the mitotic release is not shown. This should be indicated.

Figure legends for the quantification plots state 'Data are means of three independent experiments.' Do authors refer to error bars? It will indeed be powerful if the mean of each biological replicate is also shown in each plot.

Reviewer #1 (Remarks to the Author):

In this manuscript the authors identify factors required for MIDAS. MIDAS (mitotic DNA synthesis) is a recently described repair pathway to complete replication of genomic regions that remained unreplicated during interphase. The genomic regions affected by MIDAS include the common fragile sites (CFSs) that are frequently targeted in human cancers. As such, understanding MIDAS at a mechanistic level is of great interest. The authors present new data implicating PCNA ubiquitination, REV1 and the rev1-interacting region of POLD3 in MIDAS. These results are interesting, compelling and worthy of publication in Nature Communications. The authors also implicate pol-zeta in MIDAS. These results need to be better documented. It is possible that further experiments may show that pol-zeta is not implicated in MIDAS. This would be fine, as the manuscript still makes important contributions to the field and is worthy of publication.

Specific points:

1. Figs 1A-G show the establishment of cells, in which POLD1, the catalytic subunit of pol-delta, can be inducibly degraded and that degradation of POLD1 suppresses MIDAS. Figs 1H-K show that depletion by siRNA of REV3 (the catalytic subunit of pol-zeta) or REV1 (which interacts with POLD3, a subunit of the four-subunit pol-zeta polymerase: Rev3–Rev7–PolD2–PolD3) also suppresses MIDAS. These results implicate pol-delta and pol-zeta in MIDAS. A role of pol-delta in MIDAS was anticipated, whereas a role of pol-zeta was unexpected.
2. Fig. 2 shows that PCNA ubiquitination at K164 is required for MIDAS and for recruitment of Rev1 to chromatin in mitotic cells. These results suggest that PCNA K164 ubiquitination in aphidicolin-treated cells, results in the recruitment of Rev1, which, as shown in Fig. 1, is needed for MIDAS. In contrast, depletion of UBC9 does not affect MIDAS, which suggests that PCNA sumoylation is not needed for MIDAS.
3. Fig. 3 explores which is the ubiquitin ligase responsible for PCNA ubiquitination. Depletion of RAD18 suppresses aphidicolin-induced PCNA ubiquitination, whereas CDT2 depletion enhances aphidicolin-induced PCNA ubiquitination. Depletion of RAD18, but not of CDT2, also suppresses MIDAS. Thus, RAD18 is responsible for PCNA ubiquitination and MIDAS.
4. Fig. 4 examines at what stage POLD1 and REV3 function in MIDAS. The authors claim that ultrafine anaphase bridges that stain with RPA represent advanced homologous

recombination intermediates, whereas ultrafine anaphase bridges that stain with PICH represent early homologous recombination intermediates. Wild-type aphidicolin-treated cells have very few ultrafine anaphase bridges. Depletion of POLD1 leads to anaphase bridges that stain with RPA, whereas depletion of REV3 to anaphase bridges that stain with PICH. The effect of POLD1 depletion is quite strong, whereas the effect of REV3 depletion is not that strong. These results are interesting and suggest that POLD1 may have a greater role in MIDAS than REV3.

* Points 1-4 above do not require a specific response as the reviewer has provided a summary of Figures 1-4 in the manuscript.

5. Figs 5A-C show that when REV1 is depleted, then POLD1 and POLD3 are not recruited to the chromatin fraction in prometaphase cells. For reasons related to the next point, it would have been interesting to show that REV3 is also not recruited to the chromatin fraction and to also explore whether REV7 is recruited to the chromatin fraction.

* Please see our response after point 9 below.

6. Figs 5D-F show that depletion of REV7 or a compound that inhibits the interaction between REV1 and REV7 did not affect MIDAS. This observation is surprising, because the pol-zeta four-subunit polymerase consists of REV3, REV7, POLD2 and POLD3. Thus, the question arises whether the role of POLD3 in MIDAS implicates pol-zeta or pol-delta or both in MIDAS. If only pol-delta is implicated, then the findings would still be of interest, since this study clearly implicates PCNA ubiquitination, REV1 and the interaction between REV1 and POLD3 (Figs 5G-I) in MIDAS. I wonder, if the authors can resolve this issue in a revised version of the manuscript, for example by addressing the question in point 5, above, or by any other experiments they can think of (for example, using more siRNAs targeting REV3 to validate the role of REV3 in MIDAS, etc).

* Please see our response after point 9 below.

7. Figs 6A-D show that cyclin E overexpression induces MIDAS. This is a very nice and interesting result.

8. Figs 6E-I show that depleting REV1 suppresses growth (colony formation) of cyclin E overexpressing cells, but not of cells expressing normal levels of cyclin E. In contrast, REV3 depletion affects colony formation of both cyclin E overexpressing cells and cells expressing normal levels of cyclin E. It would be nice to show that MIDAS in cells overexpressing cyclin E requires REV1 and/or REV3.

- We thank the reviewer for this suggestion. We agree that this analysis would strengthen the finding that REV1/REV3 play a role in MiDAS when cells have RS induced by

oncogene activation. For this, we have carried out MiDAS analysis in U2OS cells overexpressing cyclin E. Our data indicate clearly that MiDAS induced by Cyclin E is largely abolished when either REV1 or REV3 is depleted (Supplementary Figure 11).

9. Since REV3 depletion suppresses colony formation also of cells with normal levels of cyclin E (which cells do not have MIDAS), these results again raise some questions about whether pol-zeta (REV3) has a role in MIDAS. Irrespective of whether pol-zeta has a role in MIDAS, the effect of REV1 specifically on cells overexpressing cyclin E is interesting and worth reporting.

- We appreciated the concerns from this reviewer (and reviewer #3) over the exact role of REV3 in MiDAS, particularly considering the fact that REV7 seems not to be contributing to MiDAS. To further clarify the role of REV3 in MiDAS, and the timing of its function in the MiDAS pathway, we have carried out the following new experiments:
 - 1) Using the REV3-3xFLAG plasmid (PMID: 21926160), we created a mutant REV3-3xFLAG plasmid that has the REV3 catalytic 'YGD TDS' motif changed to a catalytic dead 'YGATAS' motif (PMID: 27481099). We then performed MiDAS analysis in U2OS cells with over expression of either wild type REV3-FLAG or mutant REV3-ATA-FLAG while the endogenous REV3 was silenced using an siRNA targeting the 3'UTR of REV3 that is absent from the transfected constructs. Our results indicate that cells transfected with wild type REV3-FLAG display significantly more MiDAS than the cells transfected with REV3-ATA-FLAG mutant (Supplementary Figure 3). This is consistent with our original findings and indicates that the catalytic function of REV3 plays a role in MiDAS.
 - 2) To further clarify the order of action of REV1, REV3 and POLD1/POLD3 in the MiDAS pathway, we performed new chromatin binding assays to assess the recruitment of REV1, REV3, REV7, POLD1, POLD3 in prometaphase cells collected by mitotic shake-off. This was achieved by treating cells with siRNAs targeting REV1, REV3, or POLD3, respectively, and then collecting prometaphase cells for quantifying the chromatin-bound fraction (and the soluble protein as a control) of the relevant proteins (Figure 5A-E). Our data indicate that REV1/REV3 act before POLD3, while the binding of REV7 was not affected by the depletion of REV1, REV3, or POLD3, which is consistent with our original finding that REV7 does not appear to play a role in MiDAS.
 - 3) Unfortunately, due to the technical limitation that the level of REV3 expression is remarkably low, we could not assess the chromatin-bound fraction of REV3. This is even the case when cells are transfected by an exogenous plasmid, indicating tight regulation of REV3 levels (Sharma et al, 2012). Therefore, we could not draw a conclusion about whether REV1 or POLD3 affect the level of chromatin-bound

REV3. Despite this limitation, judging by the fact that REV3 depletion (detected by RT-qPCRs) could affect the level of chromatin-bound POLD1, POLD3 and REV1, we believe that it is appropriate to conclude that REV3 functions early in MiDAS together with REV1 and before POLD1. To illustrate that our transfection of the REV3-FLAG plasmid could indeed be detected, we performed an IP experiment using 'antiFLAG-beads' following a procedure described previously (Sharma et al, 2012), which is shown in the Appendix Figure 1 at the end of this letter.

Reviewer #2 (Remarks to the Author):

Wu et al., NCOMMS-21-35946-T

In this manuscript, Wu et al explore a role for the translesion synthesis (TLS) DNA polymerases REV1 and Rev3 in MiDAS, the replication during mitosis of fragile sites, induced by low concentrations of Aphidicolin.

Briefly, the authors do convincingly show that these TLS polymerases operate during MiDAS, dependent on PCNA ubiquitination by RAD18 and on FANCD2. Furthermore, the authors provide evidence that this process requires the interaction between REV1 and the RIR motif of POLD1, a subunit of replicative DNA polymerase delta that is also part of the REV1/REV3 complex. Finally, the authors show that inhibition of REV1 sensitizes cells to oncogene-induced replication stress.

I have mixed feelings about this manuscripts. The experiments mostly appear well executed and interpreted (but see below) and most conclusions appear warranted. However, I do have significant problems with the (lack) of novelty of the work. For example, there is an extensive body of literature on the involvement of RAD18-mediated PCNA-Ubiquitination and FANCD2 in REV1/REV3-mediated TLS but this (virtually) is not cited. E.g., a Medline search yielded 8 papers on the involvement of FANCD1/FANCD2 but none of these are cited.

- We were puzzled by this comment. The goal our manuscript was to address the hypothesis of whether TLS polymerases play a role in MiDAS considering that POLD3 is a subunit of both pol zeta and pol delta in human cells. Therefore, we have only assessed whether RAD18-mediated PCNA ubiquitination is relevant to the function of REV1 in MiDAS, and have not examined the role of FANCD2 at all. FANCD2 has been shown to localize to loci (mainly common fragile sites) from late G2 to M phase when cells are challenged with RS (PMID: 19465922; PMID: 21317883). Therefore, we have used FANCD2 as a surrogate marker to define loci undergoing RS in mitosis when performing immunofluorescence (IF) analysis with POLD1 or REV1 antibodies in the

original Figure S2 (the current Supplementary Fig. 4). Given the restrictions in space, we believe that an extensive discussion of the role of FANCD2 in TLS is not necessary, although we would be happy to add this if the reviewer/editor think this should be provided.

Also, since the factors involved in MiDAS by REV1/REV3 are essentially the same as for 'normal' REV1/REV3-dependent TLS, that also operates beyond S phase (PMID: 21908406, not cited), one wonders whether MiDAS by these TLS proteins does not represent a delayed form of 'normal' TLS. It would be interesting if this is the case by investigating whether there are differential requirements for TLS and MiDAS by REV1/REV3.

- We fully agree that it is important to clarify the timing of the action of TLS polymerases in different cell cycle phases. In the 'PMID21908406' paper, it was shown that REV1/REV3 could operate beyond S phase in G2 phase when human cells and MEFs were treated with UV-irradiation. The possible function of TLS in mitosis was not analyzed and nor was there any analysis of the response to APH-induced replication stress (RS). In our study, we have clear evidence to show that REV1 is localized to loci undergoing RS (indicated by co-IF with FANCD2) in mitosis following low-dose APH treatment in the original Figure S2 (the current Supplementary Fig. 4), which has not been reported previously to our knowledge. In addition, our results from chromatin binding assays indicate that REV1 is chromatin bound in prometaphase, particularly in cells treated with APH in S phase (Figure 5A-E), which has also not been reported to our knowledge either. Nonetheless, the PMID21908406 paper supports the notion that TLS could function beyond S phase, and we have cited this paper in the new Introduction now (Page 6, Ref. 53 in the revised manuscript). It is of course possible that the mitotic role of TLS initiates in late interphase, but this would be a very difficult to assess. We know that MiDAS requires the action of the TRAP1 ubiquitin ligase (Ref. 37 in the revised manuscript), which unloads any remaining replicative CMG helicase from chromatin as cells enter mitosis, and therefore we would contend that MiDAS in at least some cases initiates only in early prophase.
- It is also important to stress that we have investigated the differential requirements for TLS and MiDAS by silencing REV1, REV3, REV7 or Pol ϵ , respectively. Our data indicate that MiDAS does not require Pol ϵ or REV7 (Figures 5; Supplementary Figures 5 and 9), which is different from the conventional REV1/REV3-mediated TLS pathway. It is possible that we did not clarify this point previously. We have now included more discussion on this point (page 16-17).

Moreover, MiDAS has been described to require break-induced replication or symmetric fork cleavage and the authors assume but

provide no evidence that REV1/REV3-dependent MiDAS is associated with either of these breaks repair pathways (that are inappropriately extensively are described in the Discussion).

- We appreciate the reviewer's comment on this point as we could see that the original Discussion placed far too much emphasis on the current model where MiDAS is proposed to be initiated by one or two 'breaks/cleavages'. This is mainly because we wished to show that our data on REV1/REV3's role in MiDAS fit well with the different scenarios proposed in this model. We agree that we have not addressed how the function of TLS is associated to the 'break/cleavage' step in our manuscript. We believe this is beyond the scope of this manuscript and has proposed this as a future research direction in the MiDAS field. We have also reduced the amount of discussion on the MiDAS pathway to allow more discussion of the outstanding questions and future directions in general, which would be more interesting and helpful to scientists in the field we believe. (Please see more details at our response to Reviewer 3's comments.)
- We would like to point out that, in the conventional TLS pathway in S phase, the polymerase switch between REV1 and Pol delta that happens on the lagging strand is quite well understood, while the existence of any putative switch on the leading strand remains unclear (reviewed in PMID: 30442338). In the BIR model, any such switch has to occur on the leading strand. This suggests that a switch between REV1 and Pol delta in extending the leading strand as a BIR fork might be possible during the initiation of MiDAS (but would not require REV7), which is different from the conventional TLS function in S phase. Further research is also required to investigate the mechanism underlying this switch. We have modified the Discussion on this point accordingly (page 16-17).
- Our data show that REV1 depletion abolishes both the recruitment of POLD3 to mitotic chromatin and MiDAS. We believe that this is a significant piece of evidence linking REV1 with BIR (Figures 1 and 5).

Then the sensitization of cells with oncogenic replication stress by REV1 inhibition also has been described before (PMID: 32577513, not cited here).

- We apologize for overlooking this article, and thank the reviewer for pointing this out. This paper has clearly documented that TLS can restrict replication fork stalling as well as fork reversal and degradation when cells are challenged by various RS-inducing agents, including oncogene activation. Our data are complementary to those from this study considering MiDAS that is a 'rescue' pathway for cancer cells that have elevated RS caused by oncogene activation. We have included this in the new Discussion (Page 18, Ref. 91 in the revised manuscript).

In conclusion, I found the paper lacking in novelty and depth

- Based on the above discussion, the comments from Reviewers 1 and 3, and the previous and new data we have provided in the manuscript, we trust that our study has provided novel data on the function of human TLS proteins and have provided deeper understanding of mechanism of MiDAS.

Specific comments on the text:

1. Line 39: TLS is no DNA repair

- We apologize for this error and thank the reviewer for pointing it out. We have modified the text accordingly.

2. Line 92: 'the end G2 phase' should read 'the end of G2 phase'

- We apologize for this error and thank the reviewer for pointing it out. We have modified the text accordingly.

3. Line 226: CDT2 depletion unexpectedly induces PCNA Ubiquitination and a consequent increase in MiDAS (or TLS?). This unexpected finding is not at all explained. Does this imply that CDT2 acts dominantly negative over RAD18?

- We apologize for not elaborating on this unexpected finding. It was reported previously that CRL4(Cdt2) E3 ubiquitin ligase complex promotes PCNA monoubiquitylation in proliferating cells in the absence of external DNA damage, which is independent of RAD18 in human cells (Ref. 63 in the revised manuscript). Therefore, it is likely that depletion of CDT2 alone would affect the normal functioning of DNA replication, which in turn would require an increased engagement of the MiDAS pathway. Indeed, following the comment by Reviewer 3 (Rev3.1), we carried out a gammaH2AX focus analysis in mitosis in cells not exposed to exogenous RS (APH) following various siRNA treatments (Figure 7). In this figure, we show that depletion of CDT2 generates an increased level of gammaH2AX foci in the non-APH treated cells, suggesting an increased in the background level of RS, imposing a greater requirement for MiDAS. This result suggests both that CDT2 depletion cause extra RS and that it is not required for the functioning of MiDAS. On the other hand, RAD18 behaves differently in that the depletion of this protein alone (or in combination of CDT2 depletion) strongly suppresses MiDAS. We have modified the text accordingly (page 10, 17).

4. Line 282: RIR-mutant POLD3 has less MiDAS (also less TLS during G2?). But this is no direct evidence that 'REV1 recruits Pol δ to perform MiDAS via an interaction with the RIR motif of POLD3.

- We thank the reviewer for pointing this out. We have rephrased the sentence as: REV1 recruits POLD3 to perform MiDAS via an interaction with the RIR motif of POLD3 (page 13).

Comments on the experiments and Figures:

1. Graphical abstract: “Replicaltion” should read Replication.

Because a Graphical Abstract is not required for Nature Communications articles, we have removed this section now.

2. Figure 1 D: Provide FACS plots to show that Ro3306 treatment for 6 hours results in accumulation of G2 cells.

- We have now included FACS plots in Supplementary Fig. 1. Indeed, G2 cells accumulated at the end of the RO3306 treatment.

3. Figure 2 A: It is not clear which antibodies were used for this Western blot.

- We apologize for this omission. We have now included antibody information in the figure and figure legend.

4. Figure 2 G: Provide a control showing non-chromatin-bound REV1.

- We have conducted a new experiment to show this control, which is now included it in a Figure 2D.

5. Figure 2 G: What is the extra band indicated with an asterisk? This band is absent in the lane labelled with siCon.

- This is an non-specific band detected by the PCNA antibody, which is not affected by siRNAs targeting PCNA. This is shown in the new Figure 2D.

6. Figure 3 B: What is the fate of REV1 when cells are treated with siCDT2? Is there more chromatin-bound Rev1?

- As discussed above, cells treated with siCDT2 appear to encounter elevated RS in proliferating cells, but we have no evidence for a role for CDT2 in MiDAS (Figure 3). We believe, therefore, that it is beyond the scope of this manuscript to analyze the consequence of CDT2 silencing when it plays no role in the process under investigation.

7. Figure 3 E: What is the effect of RAD18 depletion on RPA-coated/PICH-coated UFBs in anaphase?

- We thank the reviewer for this suggestion. We have performed UFB analysis following RAD18 depletion with PICH and RPA antibodies in HCT116 cells treated with APH (alongside control conditions). Indeed, our new data demonstrate that there is significant increase in the frequency of PICH-coated UFBs, but not RPA-coated UFBs, following RAD18 depletion (Figure 4F-I).

8. Figure 4 A: Why did the authors treat the cells with 5-Ph-IAA for only 30 minutes, knowing that POLD1 is still detectable under these conditions (Fig 1C)?

- We were puzzled by this comment. Indeed, it takes up to 2 hours for POLD1 to be fully degraded with 5-Ph-IAA. In all relevant cases, we have treated cells with 5-Ph-IAA for 2 hours in total (1.5 hrs at the G2/M arrest time and 30 minutes after release into early mitosis; please see Figure 1D as an example). Therefore there is no discrepancy here.

9. Figures 4 C + E: Y-axis labelling is not clear. The number of RPA- positive UFBs or PICH-positive UFBs per anaphase cell can only be shown as rounded numbers.

- We apologize for the confusion. Each number is a mean of three experiments, therefore the data points are not rounded numbers. These data set was plotted using the Prism statistical software package and the source data is included in the revised submission. We have modified the figure legend accordingly.

10. Figure 5 B: Why is there more POLD3 (and not POLD1) in asynchronous cells depleted from Rev1?

- We thank the reviewer for this observation. During the revision, we found the WB analysis with asynchronous cells are variable, and therefore it is difficult to draw a definitive conclusion. In the revised manuscript, we have only focused on the prometaphase cells collected by mitotic shake-off in the new experiments (Figure 5A-E).

11. Figure 5 E: If REV1 recruits POLD1 to perform MiDAS, then treatment of cells with siRNA Rev1 should inhibit MiDAS. This is an obvious experiment that should be included in the paper.

- We were puzzled by this comment. This experiment was indeed shown in the Figure 1H-J of the original submission, which remains the same in the revised manuscript.

The experiment with the JH-RE-06 is not sufficient, since there is no positive control that this compound is able to inhibit REV1/REV7.

- We have performed a control experiment to show that this compound can inhibit REV1/REV7, as published previously (PMID: 31178121) (Supplementary Fig. 9A-C).

12. Figure 6 A: Provide FACS plots showing that cells are synchronized by the Cyclin E overexpression/Ro3306 treatment.

We have now included the requested FACS plots in Figure 6 (Figure 6B). Indeed, G2 cells were accumulated by the Cyclin E overexpression/RO3306 treatment.

Reviewer #3 (Remarks to the Author):

This manuscript reports the mechanism underlying mitotic DNA synthesis (MiDAS), which takes place in cells exposed to mild replication stress (RS), as well as in cancer cells harboring intrinsic RS. Despite its implications in cancer, the complete picture of MiDAS remains enigmatic.

It has previously been demonstrated that MiDAS is dependent on POLD3, which is involved in both Pol δ -dependent (replicative) and Pol ζ -dependent (translesion) DNA synthesis. In this study, the authors aim to determine how these polymerases act during MiDAS.

First, using an elegant AID-degron system that allows rapid depletion of POLD1, they confirmed that Pol δ indeed promotes MiDAS. They then used the siRNA-mediated depletion method to demonstrate that the REV1 and REV3 components of Pol ζ also support MiDAS. Pol η , on the other hand, appears not to be involved in MiDAS.

They also demonstrated that PCNA promotes MiDAS through the K164 residue, which is targeted for ubiquitylation by RAD18 (upon DNA damage) or CDT2 (during normal replication), or sumoylation by UBC9. Once again, using the siRNA approach, the authors demonstrated that RAD18, but not CDT2 nor UBC9, promotes MiDAS. All these observations align with the idea that stress-induced PCNA K164 mono-ubiquitylation recruits Pol ζ and potentially promotes translesion DNA synthesis in mitosis.

They then aimed to clarify how Pol ζ and Pol δ act during MiDAS. They found that POLD1 depletion increased RPA-positive UFB (considered to reflect MiDAS initiation), while REV3 depletion induced PICH-positive UFB (considered to reflect late replication intermediates, but not MiDAS initiation). Also, upon REV1 depletion, POLD3 and POLD1 recruitment to prometaphase chromatin was impaired. The depletion of REV7, which is proposed to promote Pol ζ -dependent DNA synthesis, had no impact on MiDAS. A POLD3 missense mutant, which is defective in REV1 interaction, however, failed MiDAS. Based on these observations, the authors propose that the MiDAS is initiated by PCNA K164 ubiquitylation, which in turn recruits REV3 (Pol ζ), REV1, POLD3 then POLD1 (Pol δ) in this order.

Finally, they show that U2OS cells overexpressing cyclin E, hence inducing oncogenic RS, were sensitized by the depletion of REV1 or REV3. Accordingly, they propose that the MiDAS inhibition by blocking Pol ζ -dependent translesion DNA synthesis might be an effective way to treat cancer cells.

The manuscript is well written, largely logical, and the results look very nice too. It is worth noting, however, that MiDAS events in siRNA treated cells (i.e. long-term depletion of proteins of interest) require careful assessment of the level of overall RS and associated DNA damage, especially when down-regulating factors which play significant roles during interphase. Indeed, this reviewer assumes that the increased MiDAS in siCDT2 treated cells (Fig 3E, F) reflects an increase of RS upon the depletion of CTD2, which targets a variety of proteins involved in cell cycle progression. In this context, this reviewer recommends that the authors show the degree of overall DNA stress (e.g. γ H2AX foci) in mitotically collected cells in each siRNA treated condition, ensuring the comparative assessment of the level of MiDAS relative to the level of DNA stress in mitosis (Rev3.1).

- We thank the reviewer for his/her very constructive and helpful comments. We can fully understand the concern regarding siRNA treatments, even though this remains the most widely used method to perturb the function of a protein. In response to the reviewer's excellent suggestion, we have performed a new set of experiments to assess γ H2AX foci in mitotic cells using a Quantitative image-based cytometry (QIBC) method, where DNA content and immunofluorescence staining of proteins of interest can be assessed simultaneously and automatically (also operator blind) (Figure 7). Our results indicate that, in cells not exposed to APH, the siRNA treatment alone does not generate a significant increase in the frequency of gammaH2AX foci, with the exception of cells exposed to siRNAs targeting CDT2 or Pol eta. Following APH treatment, combined with siRNA treatment, we can, as expected, see significantly increased gammaH2AX foci, although it is reassuring that the overall level of DNA damage is comparable amongst the different APH/siRNA treatments, except for that of siCDT2. This is not surprising considering that CDT2 is known to play a role in PCNA-ubiquitination even in unstressed cells, as discussed in our response to 'specific comments point 3' from Reviewer 2.

It will also be good to confirm the successful depletion of target protein in mitotically collected cells, as is shown in Fig 5B and I for chromatin-associated REV1 and POLD3, respectively.

- All of the samples collected for western blot analysis following siRNA analysis were collected at the end of G2 phase following APH and RO treatment. Considering that all of the siRNA treatments show more than 80% depletion of the target protein (except that of REV3 siRNA treatment, which is assessed by RT-qPCR), we trust that target

proteins in those cells will remain depleted during the few minutes that elapse before the cells are harvested in prometaphase. We have modified the experimental flow to illustrate the timing of WB analysis.

Determining the order of the Pol ζ and Pol δ action appears somewhat tricky. This reviewer found the interpretation of the results shown in Fig 4 highly speculative.

- Please see our response to point 9 from Reviewer 1.

Also, it is not appropriate to directly compare phenotypes of 5-Ph-IAA treated cells and siREV3 treated cells.

- We are sorry for this error and thank the review for pointing this out. We have modified the charts accordingly (Figure 4).

Fig 5 is more straightforward to interpret, although it would be good to see the level of chromatin-associated POLD1/3 upon REV3 depletion, and conversely chromatin-associated REV3 upon POLD1/3 depletion.

- Please see our response to point 9 from Reviewer 1.

Similarly, since POLD1/3 foci can be detected by IF (as shown in Fig S2 in this study; Minocherhomji et al, 2015), the authors may want to assess the levels of RS-induced mitotic POLD1/3 foci upon REV1/3 depletion to consolidate their model more directly.

- We fully agree with this suggestion. We have now performed IF analysis with a POLD1 antibody to assess POLD1 foci in mitotic cells when REV1 is depleted (Supplementary Figure 8). Our results demonstrate clearly that, upon REV1 depletion, the frequency of POLD1 foci is significantly decreased in cells following APH treatment.

The notion that Pol ζ catalytic activity may not be involved in MiDAS is intriguing. This can be tested directly by complementing REV3 depleted cells (which is defective in MiDAS) with a catalytically inactive REV3 missense mutant. This is feasible as they already have a FLAG-REV3 expression construct (Fig 1I).

- We fully agree with this suggestion, and have carried out new experiments as the reviewer suggested. Please see more details in our response to point 9 of Reviewer 1.

The reduced cell survival of cyclin E overexpressing U2OS cells upon REV1/3 depletion is also interesting, but this reviewer considers that the REV1/3-mediated TLS in interphase can explain this phenotype. The numbers of cells analyzed (n>90 in total?) in each condition are somewhat modest for the study of this sort. In the method section & the reporting summary, it

is stated that three biological replicates have been conducted, while analyses were not blinded. They should describe how foci were counted and justify their methodology.

- We apologize that we did not describe this properly in the first manuscript and Reporting Summary form. In fact, in all of the cellular assays, although we performed the cell growth and manipulation stages of the experiments without blinding the samples, for the steps involving counting foci, UFBs, or colonies, the samples were coded and analyzed blind. Also, in experiment related to mitotic cells, for each condition of each replicate, we have analyzed 30-50 cells, which is a standard requirement in this type of experiments to our knowledge. The text in the Methods section and the Reporting summary have been updated accordingly now.

Regarding the final model, this reviewer wonders if the data shown in this manuscript may explain MiDAS in the broader context beyond what the authors frame around the SSE-dependent BIR/SFC-mechanisms (Fig S7). Indeed, translesion DNA synthesis is best characterized as a mechanism that allows the DNA replication machinery to bypass DNA lesions. Hence, it seems reasonable to speculate that Pol ζ first bypasses DNA lesions generated at stalled forks, and then Pol δ takes over to carry out long-range DNA synthesis. In line with this notion, a MiDAS mechanism independent of mitotic DNA cleavage has been recently proposed (Wassing et al., Nat Comm, 2021).

Lines 123-129 (also lines 360-363)

The MiDAS model presented appears highly biased. For example, while TRAIP is shown to remove the replication machinery upon mitotic entry (mainly demonstrated in *Xenopus* and worm; and more recently in mouse ES cells, Villa et al. EMBO J, 2021), data presented in Minocherhomji et al (Fig 4a and b, Nature, 2015) indicate that many components of the replication machinery (incl. a known TRAIP substrate MCM7, as well as other MCMs, POLD1, 3 and PCNA) are retained or recruited on mitotic chromatin in cells exposed to RS in U2OS. Critically, TRAIP was identified as a PCNA binding protein (Hoffmann et al., 2015) and proposed to target PCNA itself (Feng et al. 2016). Given that the current study focuses on the role of the replication machinery and PCNA during MiDAS, a broader and critical consideration of the model in the context of different systems (experimental conditions, model organisms and cell lines) will be appropriate.

- We thank the reviewer for his/her comment on our discussion and model. We have modified the text in the Discussion in the following ways (pages 14-18):
 - 1) Reducing the discussion on the currently known factors involved in MiDAS
 - 2) Providing a more critical evaluation of the limitation of this study and raising questions for future research directions.
 - 3) Including a comparison of conventional TLS and the proposed TLS contribution in MiDAS

- 4) Including a MiDAS mechanism independent of mitotic DNA cleavage, as has been proposed recently (Wassing et al., Nat Comm, 2021; Ref. 84 in the revised manuscript).

Minor points

Line 110

SLX4-deficient cells phenotype can be explained not solely by MiDAS event, as SLX4 can resolve HR intermediates generated in interphase.

- We agree with this comment and have modified the text accordingly.

Lines 133, 136 & 140

Introduce REV1, 3 and 7 properly.

- We agree with this comment and have modified the text accordingly (page 5-6).

Fig 2

What is the band appearing in siPCNA treated cells (marked by *) in Fig 2G? The positions of molecular weight markers for PCNA blot in Fig 2A, B and G look inconsistent.

- Please see our response to point 4 of Reviewer 2. We apologize for incorrectly indicating the molecular weight of PCNA. We repeated this experiment and have included a new figure as Figure 2D.

In many Figure panels where experimental design is depicted (Figs 1D, 3D, 4A, 6A, S1D, S3A, S4A, S5D), the timing of the mitotic release is not shown. This should be indicated.

- We apologize for this omission. Mitotic release was performed after G2 arrest by washing cells with pre-warmed 1xPBS three times within 5 minutes, and then a continuation of 30 minutes with fresh new medium or with EdU for cells to progress into M phase. We have indicated this more clearly in the relevant figure legends now.

Figure legends for the quantification plots state 'Data are means of three independent experiments.' Do authors refer to error bars? It will indeed be powerful if the mean of each biological replicate is also shown in each plot.

- The error bars indicate SEM. This is explained in each figure legend in the revised manuscript.

Appendix to the rebuttal letter

Figure 1, REV3-FLAG could be detected by immunoprecipitation (IP).

Figure Legend: Western blot analysis of proteins prepared from U2OS cells transfected with a REV3-FLAG plasmid (REV3FLAG). Untransfected cells were used as a control (Con). The REV3FLAG protein was detected by an antibody against FLAG (Sigma-Aldrich; Cat# F7425,). GAPDH was used as a loading control, and detected by an antibody against GAPDH (Sigma-Aldrich; Cat#PLA0125). The bands indicated by black arrows are REV3FLAG that do not appear in the input or control IP samples. * indicates an unspecific band detected by the FLAG antibody.

Method: This experiment was carried out according to a previously published protocol (*Sharma et al, 2012*). Briefly, U2OS in one T75 cm flask (with 70% confluency) were transfected with 2 μ g REV3-FLAG plasmid (previously published; *Sharma et al, 2012*). 48 hours later, cells were collected and washed once with 1xPBS. Cell pellets were lysed with ice-cold lysis buffer (50 mM Tris HCl, pH 7.4, with 150 mM NaCl, 1 mM EDTA, and 1% Triton X-100) for 30 minutes, and the lysate was sonicated at 4°C using a water bath sonicator (The Bioruptor® Pico; Diagenode, B01060010) (setting: 30sec on / 30sec off, 10 cycles for two rounds). The cell lysate was centrifuged at 10,000 g for 20 minutes at 4°C and the supernatant was transferred to fresh tubes. 40 μ g of protein of each sample were stored as input samples, and the remaining protein (about 1.5 mg) per sample was used for IP with FLAG beads (Sigma; Cat. A2220-1ML). Beads were prepared according to the manufacture's protocol. Protein lysate of each sample was incubated with 40 μ l beads overnight at 4°C on a rotator. The beads were collected by centrifugation at 4°C, and then washed 3 times with TBS at 4°C. Protein bound to beads were eluted by incubation at 95°C for 3 minutes with 20 μ l of 2x sample buffer ((125 mM Tris HCl, pH 6.8, with 4% SDS, 20% (v/v) glycerol, and 0.004% bromphenol blue)), and centrifugation.

Reference: *Sharma S, Hicks JK, Chute CL, Brennan JR, Ahn JY, Glover TW, and Canman CE. REV1 and polymerase zeta facilitate homologous recombination repair. Nucleic Acids Res, 2012. 40(2): p. 682-91.*

REVIEWER COMMENTS

Reviewer #1 (Remarks to the Author):

The authors have addressed my concerns. I also appreciate the gH2AX experiments shown in Fig 7, to alleviate the concern that the treatments are inducing DNA damage in interphase, which might then affect MiDAS. I support publication of this revised version.

Reviewer #2 (Remarks to the Author):

See also my The authors have addressed many if not most of the concerns of the reviewers, which has led to a better manuscript.

Nevertheless, I still have significant reservations on a number of topics, most particularly as to the claims that the involvement of Rev1 and Rev3 (the catalytic subunit of PolZ) in MiDAS (replication during mitosis) mechanistically is distinct from 'classical' translesion synthesis (TLS) that operates (at least) through S and G2 phases and that shares many (or all?) of its components with MiDAS.

Below I will provide arguments for this stance and make suggestions of how to address my gripes:

1. The authors use FANCD2 as a marker for replication stress and for CFS (where MiDAS takes place).

The FANC core complex is essential for TLS by Rev1/PolZ (e.g. PMID: 22266823, PMID: 23365640, PMID: 26187992) but also for MiDAS, which would be in favour of both processes being identical (albeit during another cell cycle stage). It would be useful to investigate the involvement of the FANCD2 complex also during Rev1/PolZ-dependent MiDAS. I eluded to this in my original review, which apparently puzzled the authors.

2. Then, the authors claim that TLS and MiDAS mechanistically are distinct because TLS requires Pol eta and Rev7, but not MiDAS. This claim is not substantiated. Thus, in contrast with the author's claim, there is not such a thing as a "conventional Rev1/PolZ-mediated TLS pathway", and TLS can use other pols than Pol eta (see e.g. PMID: 30442338). Also see line 443. Thus, the finding that Pol eta is not involved in MiDAS cannot be taken as an argument that MiDAS is different from TLS.

3. The authors, in this manuscript, I feel, also do not provide convincing evidence that Rev7 is not involved in MiDAS and the lack of involvement of Rev7 in MiDAS is based on poor arguments: Rev7 is not involved in MiDAS, based on a few arguments. I have to disagree with these and am not convinced that these arguments sufficiently support the dispensability of Rev7 for MiDAS:

- An inhibitor of the Rev1/Rev7 interaction (JH-RE-06) does not affect MiDAS. This is taken as evidence that the Rev1-Rev7 interaction is not required for MiDAS. The authors have now used a control to test whether the inhibitor actually works, using cisplatin as a substrate for Rev1/Rev7-dependent TLS (Suppl. Fig. 9). But the sensitization of cells to cisplatin by the drug is only (I estimate) 30%. Indeed, cisplatin lesions may not a good substrate for Rev1 (PMID: 35819193) and therefore the drug may not be a good one to test loss of Rev1-dependent TLS. It is better to use UV radiation and, as a positive control, a Rev1

(or Rev7)-deficient cell line.

- The authors show that Rev7 binds to chromatin in the absence of Rev1, Rev3 or PolD3. This also is taken as evidence that Rev7 is not involved in MiDAS. But Rev7 has Rev1/3-independent functions that may operate at chromatin (such as in homologous recombination, PMID: 33051298, and in dechromatinization, PMID: 28330620). So Rev7 binding at chromatin is not predictive for absence of a role in MiDAS. Why was MiDAS not investigated in Rev7-deficient cells, which would be a much better experiment?

The question whether Rev1/Rev3-dependent MiDAS (during mitosis) represents the same pathway as TLS (during S phase), or a mechanistically distinct entity, may be addressed by performing some (or all) of the experiments performed in this manuscript on mitotic cells, also on S phase cells.

Issues with respect to the rebuttal on my review report (reviewer 2):

Rebuttal: “We would like to point out that, in the conventional TLS pathway in S phase, the polymerase switch between REV1 and Pol delta that happens on the lagging strand is quite well understood, while the existence of any putative switch on the leading strand remains unclear (reviewed in PMID: 30442338).” First, there is in my opinion, no evidence for a handover of TLS from Rev1/PolZ to PolD. This would be illogical as TLS entails gap (i.e. Okazaki fragment) filling at the lagging strand, and has no need for processive replication. And on the leading strand, the situation is similar although here processive replication may be re primed by PrimPol in most cases (PMID: 34624216, PMID: 34508659, PMID: 34186497).

Rebuttal: “Our data show that REV1 depletion abolishes both the recruitment of POLD3 to mitotic chromatin and MiDAS. We believe that this is a significant piece of evidence linking REV1 with BIR (Figures 1 and 5).” This would be only true if BIR is the only MiDAS pathway. This may not be true as there may be BIR-dependent and BIR-independent (TLS) pathways for MiDAS.

Rebuttal 2 (comments section on the experiments and Figures). RO-3306 is used to synchronize cells in M phase. I am sorry but I am not impressed by the G2 arrest provided by the drug, the cells are largely in G1 and S phases (Fig. 6B and S1 and left panel below as included in the review attachment). How sure are the authors that the cells studied after release from the drug really are all in mitosis? Below (right panel in the review attachment), please see an example of a strong G2 arrest induced by RO-3306 following labeling with BrdU during S phase (ploidy on X axis, BrdU on Y axis).

Other issues:

1. Line 130: “This enzyme (Rev3/Rev7) is believed to be the main polymerase that can bypass a wide range of lesions” Incorrect, Rev3/Rev7 does rarely bypass lesions, it does perform extension from a misaligned primer. A similar mis-statement is made in line 403.

2. Line 142: “REV1 and Pol ζ operate in G2 to counteract DNA damage induced by UV radiation⁵³.”

REV1/POLZ do not counteract DNA damage, they aid in replication of damaged DNA.

3. Line 176: "We observed that MiDAS was also largely abolished when either REV3 or REV1 was depleted, indicating that TLS polymerases also play a role in MiDAS" Since Rev3-deficient cells are extremely sensitive to replication trouble, has it been ascertained that the Rev3KD cells do enter mitosis (rather than staying arrested in S/G2)?

4. Line 316. It is suggested that PolD3 regulates Rev1/Rev3 during MIDAS via its RIR domain, alternative to Rev7. But PolD3 is downstream of Rev1/3, as suggested by this manuscript, whereas Rev7 (presumably) is upstream of Rev1/Rev3. So is it feasible than that the PolD3 interaction can functionally complement for the Rev7 interaction?

5. Line 342: Cyclin E-overexpressing cells are sensitized by Rev1 depletion. This is taken as an argument that MiDAS protects sensitizes cells with oncogene activation. However, the dominant phenotype of Rev1 loss is a TLS defect, and that also sensitizes oncogene-activated cells (see Ref. 91, PMID: 32577513). How reasonable then is it to assume that the phenotype observed here results from loss of MiDAS rather than from loss of TLS?

6. Line 442 (also see above): "Interestingly, recently, it was shown that a small molecule inhibitor targeting the C-terminal domain of REV1 could synergize with replication-gap inducing cancer treatments" This is not an appropriate attribution to ref 91 (PMID: 32577513); these authors have investigated the role of Rev1 in surviving oncogenic replication stress (and not only "replication gap-inducing treatments"), precisely the same experiment as described in the current manuscript.

Reviewer #3 (Remarks to the Author):

The manuscript has improved significantly and I am satisfied with their response. Below, I list a small number of suggested corrections, which should be considered before proceeding.

Line 66: I guess the authors meant "Only when additional genetic or epigenetic changes occur (e.g. loss of the p53 pathway), the effect of the DDR pathway compromised, permitting tumorigenesis to proceed." (remove "is" after the bracket)

Line 122: human "replicative" DNA polymerase δ (add "replicative" to be explicit)

Line 135: define "Pol ζ " (no explanation as far as I could see - Pol ζ with two subunits?)

Line 145: define "POLD1" (first time describing "POLD1")

Line 214: newly-borne -> newly-bone?

Line 357: when discussing RAD51-dependent MiDAS events, also consider citing Mocanu et al., Cell Rep 2022 (PMID: 35443178). This paper proposes the RAD51- and RAD52-dependent 'S-to-M DNA synthesis runover' model.

REVIEWER COMMENTS and response from the authors (in blue)

Reviewer #1 (Remarks to the Author):

The authors have addressed my concerns. I also appreciate the gH2AX experiments shown in Fig 7, to alleviate the concern that the treatments are inducing DNA damage in interphase, which might then affect MiDAS. I support publication of this revised version.

Reviewer #2 (Remarks to the Author):

See also my The authors have addressed many if not most of the concerns of the reviewers, which has led to a better manuscript.

Nevertheless, I still have significant reservations on a number of topics, most particularly as to the claims that the involvement of Rev1 and Rev3 (the catalytic subunit of PolZ) in MiDAS (replication during mitosis) mechanistically is distinct from 'classical' translesion synthesis (TLS) that operates (at least) through S and G2 phases and that shares many (or all?) of its components with MiDAS.

Below I will provide arguments for this stance and make suggestions of how to address my gripes:

1. The authors use FANCD2 as a marker for replication stress and for CFS (where MiDAS takes place). The FANC core complex is essential for TLS by Rev1/PolZ (e.g. PMID: 22266823, PMID: 23365640, PMID: 26187992) but also for MiDAS, which would be in favour of both processes being identical (albeit during another cell cycle stage). It would be useful to investigate the involvement of the FANCI/D2 complex also during Rev1/PolZ-dependent MiDAS. I eluded to this in my original review, which apparently puzzled the authors.

* We are aware of the fact that FANC core complex can regulate REV1 when cells are challenged with DNA damages including MMC, hydroxyurea (HU), camptothecin (CPT) (PMID: 22266823), UV or laser treatment (PMID: 23365640, PMID: 26187992). However, to our knowledge, it remains unclear whether FANCD2/I play a direct role in MiDAS when cells were treated with low dose of APH. In our last rebuttal letter, we pointed out the analysis of FANCD2/I is beyond the scope of this study as this manuscript is focused on the identification of polymerases involved in MiDAS. Therefore, we could not understand

why the analysis of FANCD2 was necessary. Following the reviewer's new comment, we performed MiDAS analysis on a cell line that is deficient in FANCD2. We observed that MiDAS is not affected in this cell line when they are treated with APH. This suggests that, although FANCD2 is recruited to loci undergoing replication stress from S/G2 phase to M phase when cells are challenged with low dose APH, it does not play a direct role in MiDAS. Further investigation is required to dissect if other FANCD2 associated factors play a role in MiDAS. For example, the FAAP20 protein is a good candidate as it has been reported to provide a link between the Fanconi anemia pathway and TLS polymerase activity when cells were treated with MMC, HU, or CPT (PMID: 22266823). The work in this direction is novel and is certainly beyond the scope of this manuscript.

2. Then, the authors claim that TLS and MiDAS mechanistically are distinct because TLS requires Pol eta and Rev7, but not MiDAS. This claim is not substantiated. Thus, in contrast with the author's claim, there is not such a thing as a "conventional Rev1/PolZ-mediated TLS pathway", and TLS can use other pols than Pol eta (see e.g. PMID: 30442338). Also see line 443. Thus, the finding that Pol eta is not involved in MiDAS cannot be taken as an argument that MiDAS is different from TLS.

* We thank the reviewer for pointing this out. We tested Pol eta because Pol eta was known to play a role in fragile site maintenance (PMID: 23609533) and cooperates with Pol Zeta when bypassing of cisplatin lesions (PMID: 24449906). We realize that the original text was an overstatement since we have only tested the relevance of Pol zeta and Pol eta to MiDAS. We have modified the text accordingly (in the Result related to Supplementary Figure 5 and Discussion).

3. The authors, in this manuscript, I feel, also do not provide convincing evidence that Rev7 is not involved in MiDAS and the lack of involvement of Rev7 in MiDAS is based on poor arguments: Rev7 is not involved in MiDAS, based on a few arguments. I have to disagree with these and am not convinced that these arguments sufficiently support the dispensability of Rev7 for MiDAS:

- An inhibitor of the Rev1/Rev7 interaction (JH-RE-06) does not affect MiDAS. This is

taken as evidence that the Rev1-Rev7 interaction is not required for MiDAS. The authors have now used a control to test whether the inhibitor actually works, using cisplatin as a substrate for Rev1/Rev7-dependent TLS (Suppl. Fig. 9). But the sensitization of cells to cisplatin by the drug is only (I estimate) 30%. Indeed, cisplatin lesions may not a good substrate for Rev1 (PMID: 35819193) and therefore the drug may not be a good one to test loss of Rev1-dependent TLS. It is better to use UV radiation and, as a positive control, a Rev1 (or Rev7)-deficient cell line.

- The authors show that Rev7 binds to chromatin in the absence of Rev1, Rev3 or PolD3. This also is taken as evidence that Rev7 is not involved in MiDAS. But Rev7 has Rev1/3-independent functions that may operate at chromatin (such as in homologous recombination, PMID: 33051298, and in dechromatinization, PMID: 28330620). So Rev7 binding at chromatin is not predictive for absence of a role in MiDAS. Why was MiDAS not investigated in Rev7-deficient cells, which would be a much better experiment?

*We appreciated the reviewer's concern of whether REV7 is playing a role in MiDAS. In the previous submission (July 2022), three pieces of evidence were included to support this conclusion:

- 1) The chromatin bound REV7 in prometaphase cells (collected by 'mitotic shake-off') was not affected by KD of REV3, REV1 or POLD3 (Figure 5)
- 2) siRNA knockdown of REV7 does not affect MiDAS (the previous Supplementary Figure 9D-G)
- 3) The inhibitor to REV1-REV7 interaction does not affect MiDAS (the previous Supplementary Figure 9)

It seems that the reviewer had overlooked our data related to MiDAS analysis following siRNA knockdown of REV7. Nonetheless, we fully agree that it is better to use a REV7 knockout cell line to test whether MiDAS is affected. We therefore obtained a pair of isogenic U2OS cells with or without REV7 (from Prof. A. D'Andrea' group; Pubmed 31915374) and performed MiDAS analysis on them. Our data demonstrate that, consistent with our previous data, loss of REV7 has no effect on MiDAS (the new Supplementary Fig. 9e-h). Because of this new evidence, the previous data related to the inhibitor of

Rev1/Rev7 interaction (JH-RE-06) became irrelevant. Therefore, we have removed these data from this figure accordingly.

The question whether Rev1/Rev3-dependent MiDAS (during mitosis) represents the same pathway as TLS (during S phase), or a mechanistically distinct entity, may be addressed by performing some (or all) of the experiments performed in this manuscript on mitotic cells, also on S phase cells.

*We appreciated the reviewer's comment on this point. This comment is also in agreement of a question remains to be clarified in the MiDAS field: Is MiDAS a continuation of DNA synthesis starts in S/G2 phase, or *de novo* starting in early mitosis (selected PMIDs: 26633632; 34508092;35443178)? Based on the evidence published so far, it is possible that MiDAS is a phenomenon containing DNA synthesis of both of these two scenarios. In view of this, we agree that it is possible that TLS could contributing to MiDAS from G2 to M phase. We have therefore modified the text in the Discussion accordingly. To pinpoint the exact starting time of DNA synthesis, a different set of analysis would be required. This could include examining the replication profiles in S phase, G2 phase, and M-phase by 'Repli-seq', or DNA fiber assays on loci that are known to be vulnerable to APH treatment (eg common fragile sites), which can be achieved by combining DNA fiber assay with Fluorescence *in Situ* Hybridization (FISH). This direction is very interesting and would be worth pursuing in the future, and is beyond the scope of this study.

Issues with respect to the rebuttal on my review report (reviewer 2):

Rebuttal: "We would like to point out that, in the conventional TLS pathway in S phase, the polymerase switch between REV1 and Pol delta that happens on the lagging strand is quite well understood, while the existence of any putative switch on the leading strand remains unclear (reviewed in PMID: 30442338)." First, there is in my opinion, no evidence for a handover of TLS from Rev1/PoiZ to PoiD. This would be illogical as TLS entails gap (i.e. Okazaki fragment) filling at the lagging strand, and has no need for processive replication. And on the leading strand, the situation is similar although here processive

replication may be re primed by PrimPol in most cases (PMID: 34624216, PMID: 34508659, PMID: 34186497).

*We would like to clarify that our previous statement on this point was cited from a published review (PMID: 30442338). Following the reviewer's comment on this point, and by looking into literature in more detail, we came to the view that any differences between the exact function of TLS in mitosis and in S phase will only be clarified by further analyses. Therefore, we have removed this speculation from the Discussion.

Rebuttal: "Our data show that REV1 depletion abolishes both the recruitment of POLD3 to mitotic chromatin and MiDAS. We believe that this is a significant piece of evidence linking REV1 with BIR (Figures 1 and 5)." This would be only true if BIR is the only MiDAS pathway. This may not be true as there may be BIR-dependent and BIR-independent (TLS) pathways for MiDAS.

*We fully agree that MiDAS is a process that contains several pathways. We have actually illustrated this point in our last revised Discussion (Lines 352 to 365), and the Supplementary Figure 12, where two models of the pathways leading to MiDAS, namely 'BIR and SFC', were presented. In our last rebuttal letter, this statement was meant to say 'We believe that this is a significant piece of evidence linking REV1 with BIR (Figures 1 and 5), which is one of the pathways involved in MiDAS.'

Rebuttal 2 (comments section on the experiments and Figures). RO-3306 is used to synchronize cells in M phase. I am sorry but I am not impressed by the G2 arrest provided by the drug, the cells are largely in G1 and S phases (Fig. 6B and S1 and left panel below as included in the review attachment). How sure are the authors that the cells studied after release from the drug really are all in mitosis? Below (right panel in the review attachment), please see an example of a strong G2 arrest induced by RO-3306 following labeling with BrdU during S phase (ploidy on X axis, BrdU on Y axis).

*We would like to clarify that RO-3306 was used to enrich G2 phase cells, but not to synchronize cells. The aim of this treatment was simply to harvest enough mitotic cells for

MiDAS analysis. To avoid any further confusion, in all of the flowchart of the figures submitted this time, 'G2 phase arrest' is changed to 'G2 phase enrichment' in the corresponding panels.

In addition, in the cellular biology field, it is a quite well-known phenomenon that adherent mammalian cells are flat when they are in interphase but round up into a spherical structure when they are undergoing mitosis. Therefore a method of 'mitotic shake-off' has been developed to collect mitotic cells

(<https://www.oxfordreference.com/view/10.1093/acref/9780199549351.001.0001/acref-9780199549351-e-6109>; A Dictionary of Biomedicine, 2010, by John Lackie). To demonstrate that our protocol has indeed helped to enrich and harvest mitotic cells, we have provided both brightfield and IF image analysis of cells before and after being harvested by the 'mitotic shake-off' step as part of our protocol of MiDAS analysis (Appendix Figure 2).

Other issues:

1. Line 130: "This enzyme (Rev3/Rev7) is believed to be the main polymerase that can bypass a wide range of lesions" Incorrect, Rev3/Rev7 does rarely bypass lesions, it does perform extension from a misaligned primer. A similar mis-statement is made in line 403.

*We appreciated this comment and have modified relevant text accordingly.

2. Line 142: "REV1 and Pol ζ operate in G2 to counteract DNA damage induced by UV radiation⁵³." REV1/POLZ do not counteract DNA damage, they aid in replication of damaged DNA.

*We agree with this comment and have modified relevant text accordingly.

3. Line 176: "We observed that MiDAS was also largely abolished when either REV3 or REV1 was depleted, indicating that TLS polymerases also play a role in MiDAS" Since Rev3-deficient cells are extremely sensitive to replication trouble, has it been ascertained that the Rev3KD cells do enter mitosis (rather than staying arrested in S/G2)?

*Yes, we can confirm that cells do enter mitosis with REV3 siRNA treatment in our analysis. This was because we used low dose of APH that only causes mild replication stress. Also, there would be still some active REV3 in the siRNA treated cells, as siRNA treatment cannot deplete the expression of REV3 totally.

4. Line 316. It is suggested that PolD3 regulates Rev1/Rev3 during MIDAS via its RIR domain, alternative to Rev7. But PolD3 is downstream of Rev1/3, as suggested by this manuscript, whereas Rev7 (presumably) is upstream of Rev1/Rev3. So is it feasible than that the PolD3 interaction can functionally complement for the Rev7 interaction?

We thank reviewer for pointing this out. We would like to clarify that we have meant to suggest POLD1 works downstream of TLS, but not POLD3, since it's a subunit of both Pol zeta and pol delta. In fact, consistent with this proposal, in our last model shown in Suppl. Fig. 12B, we have indicated that POLD3 binds to REV1 in the G2 phase and mitosis, upstream of POLD1 action in MiDAS. We have modified text in the Result section related to Fig.5 now.

5. Line 342: Cyclin E-overexpressing cells are sensitized by Rev1 depletion. This is taken as an argument that MiDAS protects sensitizes cells with oncogene activation. However, the dominant phenotype of Rev1 loss is a TLS defect, and that also sensitizes oncogene-activated cells (see Ref. 91, PMID: 32577513). How reasonable then is it to assume that the phenotype observed here results from loss of MiDAS rather than from loss of TLS?

* We appreciate the reviewer's comment on this. Our conclusion of that 'inhibition of REV1 could sensitize cells to oncogene activation via inhibition of MiDAS' was based on the fact that: 1) U2Os was found as a TLS independent cells (please see Figure 4 of PMID: 32577513); 2) upon oncogene activation and REV1/REV3 KD, MiDAS was significantly reduced in these cells, which could lead to the genome instability and more cell death (Supplementary figure 11, which was included in the last revision following Reviewer 1's suggestion). Interestingly, the same U2OS cell line with an inducible

oncogene (Tet off) system was also analyzed in the Figure 3 of PMID 32577513. This figure was used to show that TLS can overcome oncogene-induced stress response and promote cell fitness. We noticed that these cells were treated with 24 hours with 'Tet off' to induce Cyclin E overexpression, and the amount of Cyclin E expression was very low judging by the WB in this assay. From our experience, it takes 7 days for these cells to expression cyclin E when they are cultured with media containing no doxycycline. In our assays with this cell line, we treated our cells for 14 days to induce much more pronounced oncogene overexpression (please see our Fig. 6 and Supplementary Fig. 11). Therefore, it is difficult to know whether the phenotype observed in PMID: 32577513 was not due to loss of MiDAS either.

Ideally, it is best to analyze this cell line with TLS enzymes that can be degraded specifically at S phase or M phase, i.e. a REV1 or REV3 degron system. In addition, because U2OS cell line is already a cancer cell line that has genomic changes to cope with its intrinsic replication stress, in the future, it would be necessary to analyze the function of oncogene activation in a non-cancer cell line. All of these future directions are beyond the scope of this study. We have modified text in Result relevant to Figure 6 accordingly.

6. Line 442 (also see above): "Interestingly, recently, it was shown that a small molecule inhibitor targeting the C-terminal domain of REV1 could synergize with replication-gap inducing cancer treatments" This is not an appropriate attribution to ref 91 (PMID: 32577513); these authors have investigated the role of Rev1 in surviving oncogenic replication stress (and not only "replication gap-inducing treatments"), precisely the same experiment as described in the current manuscript.

It is clear to us that the PMID32577513 paper has shown that activation of translesion synthesis (TLS) polymerases suppresses the accumulation of replication gaps during replication stress (Figs 1, 2 of this paper). Those authors also demonstrated that REV1 inhibition (TLSi) synergizes with gap-inducing therapies including inhibitors of ATR or Wee1 (Fig. 5 of this paper). For clarity, the Title page and Abstract of this paper is copied as below:

Inhibition of the translesion synthesis polymerase REV1 exploits replication gaps as a cancer vulnerability

Sumeet Nayak, Jennifer A Calvo, Ke Cong, Min Peng, Emily Berthiaume, Jessica Jackson, Angela M Zaino, Alessandro Vindigni, M Kyle Hadden, Sharon B Cantor. PMID: 32577513

Abstract

The replication stress response, which serves as an anticancer barrier, is activated not only by DNA damage and replication obstacles but also oncogenes, thus obscuring how cancer evolves. Here, we identify that oncogene expression, similar to other replication stress-inducing agents, induces single-stranded DNA (ssDNA) gaps that reduce cell fitness. DNA fiber analysis and electron microscopy reveal that activation of translesion synthesis (TLS) polymerases restricts replication fork slowing, reversal, and fork degradation without inducing replication gaps despite the continuation of replication during stress. Consistent with gap suppression (GS) being fundamental to cancer, we demonstrate that a small-molecule inhibitor targeting the TLS factor REV1 not only disrupts DNA replication and cancer cell fitness but also synergizes with gap-inducing therapies such as inhibitors of ATR or Wee1. Our work illuminates that GS during replication is critical for cancer cell fitness and therefore a targetable vulnerability.

Reviewer #3 (Remarks to the Author):

The manuscript has improved significantly and I am satisfied with their response. Below, I list a small number of suggested corrections, which should be considered before proceeding.

Line 66: I guess the authors meant "Only when additional genetic or epigenetic changes occur (e.g. loss of the p53 pathway), the effect of the DDR pathway compromised, permitting tumorigenesis to proceed." (remove "is" after the bracket)

*We agree with this comment and have modified relevant text accordingly.

Line 122: human "replicative" DNA polymerase δ (add "replicative" to be explicit)

*We agree with this comment and have modified relevant text accordingly.

Line 135: define "Pol ζ " (no explanation as far as I could see - Pol ζ with two subunits?)

*We agree with this comment and have modified relevant text accordingly.

Line 145: define "POLD1" (first time describing "POLD1")

*We agree with this comment and have modified relevant text accordingly.

Line 214: newly-borne -> newly-bone?

*This should be 'newly-born'. We have modified relevant text accordingly.

Line 357: when discussing RAD51-dependent MiDAS events, also consider citing Mocanu et al., Cell Rep 2022 (PMID: 35443178). This paper proposes the RAD51- and RAD52-dependent 'S-to-M DNA synthesis runover' model.

*We agree with this comment, which is also mentioned in our response to Reviewer 2. We have modified relevant text in Discussion accordingly.

A**B****C****D**
Appendix Figure 1: FANCD2 does not play a direct role in MiDAS

A) An immortalized fibroblast cell line derived from a Fanconi Anemia patient with loss of FANCD2 function (PD20i) was analyzed to address whether FANCD2 plays a role in MiDAS (PMID: 11239453; PMID: 8782494) (a kind gift from Dr A. D’Andrea’s group). These cells were treated with APH (0.4 μM for 16 hours (h) to induce replication stress. To enrich mitotic cells, asynchronous cells were treated with CDK1 inhibitor RO-3306 (7 μM; APEXBio) in the last 6 hours of APH treatment. Cells were then rinsed with pre-warmed medium (37°C) three times (within 5 minutes) before being released into pre-warmed EdU-containing cell culture media to allow them to progress into mitosis. Prometaphase cells were then collected by mitotic-shake off, re-seeded onto Poly-L-Lysine-coated slides (Sigma Aldrich) for imaging analysis. **B)** Western blot analysis of FANCD2 in the PD20i cell line. 50 μg of protein from whole cell lysate was loaded in each sample. The whole cell lysate from a human fibroblast cell line GM05848 (Coriel Institute) was used as a positive control for FANCD2 expression. Tubulin was used as a loading control. **C)** Representative immunofluorescence images and quantification **(D)** of MiDAS foci (labelled with EdU; red) in prometaphase cells. DNA was stained with DAPI (blue) . Scale bars, 10 μm. 45 mitotic cells were analysed in each condition of each experiment. Data in the chart are means of three independent experiments and plotted with Prism. Error bars represent SEM. P values were calculated using a two-tailed non-parametric Mann–Whitney test. ****: p<0.0001.

Appendix Figure 2: Mitotic cells enriched by Ro3306 treatment and mitotic shake-off.

A) Representative brightfield microscopy images of U2OS cells following a treatment of 16-hour APH (0.4 μ M) and RO3306 (7 μ M in the last 6 hours), which is indicated as 'RO3306 0 min'; cells were then released into pre-warmed cell culture media for 30 minutes to allow them to progress into mitosis, which is indicated as 'RO3306 30 min'; prometaphase cells were then collected by mitotic-shake off, re-seeded onto Poly-L-Lysine-coated slides (Sigma Aldrich) for imaging analysis, which is indicated as 'RO3306 30 min + mitotic shake-off'. The lower panel shows zoomed images of regions highlighted in the yellow boxes in the corresponding upper panel. Cells are flat during interphase but round up during mitosis, and acquire a bright and refractile appearance. **B)** Representative immunofluorescence images of U2OS cells at stages of 'RO3306 0 min', 'RO3306 30 min', 'RO3306 30 min + mitotic shake-off' as described in panel A. These cells were fixed and then incubated with antibody against phosphorylated Histone H3 (Ser10) (pHisH3; green) to distinguish mitotic cells. DNA was stained with DAPI (blue). Scale bars, 500 μ m. Mitotic cells are positive with pHisH3 (green) and lack a nuclear membrane, and are characterized by a rugged and irregular shaped nuclear DNA mass.

REVIEWERS' COMMENTS

Reviewer #2 (Remarks to the Author):

The authors have satisfactorily addressed most my pertaining comments and concerns on revision 1 of their manuscript. It is a better paper now and I find it acceptable for publication.